

# European pollen-based REVEALS land-cover reconstructions for the Holocene: methodology, mapping and potentials

Esther Githumbi[1,2], Ralph Fyfe[3], Marie-Jose Gaillard[2], Anna-Kari Trondman[2,4], Florence Mazier[5], Anne-Birgitte Nielsen[6], Anneli Poska[1,7], Shinya Sugita[8], Martin Theuerkauf[9], Jessie Woodbridge[3], Julien Azuara[10], Angelica Feurdean[11,12], Roxana Grindean[12], Vincent Lebreton[10], Laurent Marquer[13], Nathalie Nebout-Combourieu[10], Migle Stancikaite[14], Ioan Tanţău[12], Spassimir Tonkov[15], Lyudmila Shumilovskikh[16], and LandClimII data contributors[17+].

[1]Department of Physical Geography and Ecosystem Science, University of Lund, 22362 Lund, Sweden
[2]Department of Biology and Environmental Science, Linnaeus University, 39182 Kalmar, Sweden
[3]School of Geography, Earth and Environmental Sciences, University of Plymouth, PL4 8AA Plymouth, United Kingdom
[4]Division of Education Affairs, Swedish University of Agricultural Science (SLU), 23456 Alnarp, Sweden
[5]Environmental Geography Laboratory, GEODE UMR 5602 CNRS, Université de Toulouse Jean Jaurès, 31058 Toulouse, France
[6]Department of Geology, Lund University, 22100 Lund, Sweden
[7]Department of Geology, Tallinn University of Technology, 19086 Tallinn, Estonia
[8]Institute of Ecology, Tallinn University of Technology, 10120 Tallinn, Estonia
[9]Institute of Botany and Landscape Ecology, EMAU Greifswald, 1748 Greiswald, Germany
[10]Département Homme et Environnement, UMR 7194 Histoire Naturelle de l'Homme Préhistorique, 75013 Paris, France
[11]Senckenberg Biodiversity and Climate Research Centre (BiK-F), 60325 Frankfurt am Main, Germany
[12]Department of Geology, Faculty of Biology and Geology, Babeş-Bolyai University, 400084 Cluj-Napoca, Romania
[13]Department of Botany, University of Innsbruck, 6020 Innsbruck, Austria
[14]Institute of Geology and Geography, Vilnius University, Vilnius, LT-03101 Vilnius, Lithuania
[15]Department of Botany, Sofia University St. Kliment Ohridski, 1164 Sofia, Bulgaria
[16]Department of Palynology and Climate Dynamics, Georg-August-University, 37073 Göttingen, Germany
[17]+Team list
+A full list of authors appears at the end of the paper.

*Correspondence to*: Esther Githumbi (esther.githumbi@lnu.se)



**Abstract.** Quantitative reconstructions of past land-cover are necessary for research into the processes involved in climate-human-land interactions. We present the first temporally continuous pollen-based land-cover reconstruction for Europe over the Holocene (last 11,700 cal yr BP). We describe how vegetation cover has been quantified from pollen records at a 1°x1° spatial scale using the 'Regional Estimates of VEgetation Abundance from Large Sites' (REVEALS) model. REVEALS has been applied to 1128 pollen records across Europe and part of the Eastern Mediterranean-Black Sea-Caspian-Corridor (30°-75°N, 25°W-50°E) to reconstruct the cover of 31 plant taxa assigned to 12 plant functional types (PFTs) and three land-cover types (LCTs). A new synthesis of relative pollen productivities (RPPs) available for European plant taxa was performed for this reconstruction. It includes > 1 RPP values for 39 taxa, and single values for 15 taxa (total of 54 taxa). As an illustration, we present maps of the results for five taxa (*Calluna vulgaris,* Cerealia-t*, Picea abies, Quercus* deciduous and *Quercus* evergreen) and three LCTs (open land (OL), evergreen trees (ET) and summer-green trees (ST)) for 8 selected time windows. We discuss the reliability of the REVEALS reconstructions and issues related to the interpretation of the results in terms of landscape openness and human-induced vegetation change. We then describe the current use of this reconstruction and its future potential utility and development. The REVEALS data presented here can be downloaded from https://doi.pangaea.de/10.1594/PANGAEA.937075?format=html#download.



## 1 Introduction

The reconstruction of past land cover at global, continental and sub-continental scales is necessary for the evaluation of climate models, land-use scenarios and the study of past climate – land cover interactions. Vegetation plays a significant role within the climate system through biogeochemical and biogeophysical feedbacks and forcings (Foley, 2005; Gaillard et al., 2010, 2015, 2018; Strandberg et al., 2014). Land use has modified the land cover of Europe over Holocene timescales at local, regional and continental scales (e.g. Roberts et al., 2019; Trondman et al., 2015; Woodbridge et al., 2014). Concerted efforts have been made to model land-use and land-cover change (LULCC) over Holocene time scales (e.g. HYDE 3.2 (Klein Goldewijk et al., 2017) and KK10 (Kaplan et al., 2011)). KK10 has been used to assess the impact of the scale of deforestation between 6000 and 200 cal yr BP in Europe on the regional climate in the climate modelling study of Strandberg et al (2014). The KK10-inferred land-cover change resulted in cooling or warming of the regional climate by 1º to 2º depending on the season (winter or summer) and/or geographical location. Major changes in the forest cover of Europe over the Holocene may therefore have had a significant impact on past regional climate, particularly those driven by deforestation since the start of agriculture. Estimating past land-cover change can enable quantification of the scale at which human impact on terrestrial ecosystems perturbed the climate system. This in turn allows us to consider when environmental changes moved beyond the envelope of natural variability (Ruddiman, 2003; Ruddiman et al., 2016). We focus here on the role of LULCC in the climate system; anthropogenic land-cover change can have broader consequences on other processes and changes, such as erosion and fluvial systems (Downs and Piégay, 2019), biodiversity loss (Barnosky et al., 2012), nutrient cycling (Guiry et al., 2018; McLauchlan et al., 2013), habitat exploitation by megafauna (Hofman-Kamińska et al., 2019) and wider ecosystem functioning (Ellis, 2015; Stephens et al., 2019).

The Earth System Modelling (ESM) community use LULCC model scenarios, along with dynamic vegetation models, to understand interactions between different components of the earth system in the past (e.g. Gilgen et al., 2019; Smith et al., 2016; He et al., 2014; Hibbard et al., 2010). Disagreement between LULCC scenarios suggests that their evaluation is needed using independent, empirical datasets (Gaillard et al., 2010). Pollen-based reconstruction of past land cover represents probably the best empirical data for this purpose, as a direct proxy for vegetation, and for the ubiquity of data across the continent of Europe (e.g. Gaillard et al., 2010, 2018). Moreover, the 'Regional Estimates of VEgetation Abundance from Large Sites' (REVEALS) model developed by Sugita (2007a) makes it possible to quantify plant cover from pollen records at a regional spatial scale of ca. 100 km x 100 km (e.g. Hellman et al., 2008a). The first pollen-based REVEALS reconstruction of plant cover over the Holocene covering a large part of Europe (Trondman et al., 2015) was used for the assessment of LULCC scenarios (Kaplan et al., 2017), and helped to evaluate climate model simulations using LULCC scenarios (Strandberg et al., 2014). A comparison between REVEALS-based open land cover from pollen records and Holocene deforestation as simulated by HYDE 3.1 and KK10 showed that the REVEALS reconstructions were more similar to the KK10 scenarios than the HYDE 3.1 ones (Kaplan et al., 2017). Therefore, estimates of past plant cover from pollen are essential to both test and constrain LULCC models, and also provide alternative inputs to Earth System Models (ESMs), Regional Climate Models (RCMs) and



ecosystem models (Gaillard et al., 2018; Harrison et al., 2020). This allows improved assessments of biogeophysical and
biogeochemical forcings on climate due to LULCC over the Holocene (Gaillard et al., 2010; Harrison et al., 2020; Ruddiman
et al., 2016; Strandberg et al., 2014).
Europe is of particular interest as one of the regions of the globe that has experienced major human-induced land-cover
transformations. Europe has large N-S and W-E gradients in modern and historical climate and land use (Marquer et al., 2014,
2017). Early agriculture dates from the start of the Holocene in the SE Mediterranean region (Roberts et al., 2019; Shennan,
2018), and human impact on vegetation across most of Europe is characterized by early land-cover changes through agriculture
(e.g. Marquer et al., 2014; Trondman et al., 2015). There is therefore a clear need to extend quantitative vegetation
reconstruction to the whole of Europe, including for the first time the Mediterranean region and additional areas of Eastern
Europe, and to cover the whole Holocene to capture transient vegetation change at sub-millennial time scales. Europe has a
deep history of pollen data production (Edwards et al., 2017) and an open-access repository for pollen records (the European
Pollen Database (EPD): (Fyfe et al., 2009)), resulting in abundant pollen records that can be used for data-driven
reconstructions of past vegetation patterns at continental scales. Vegetation reconstructions for Europe based on pollen data
have used community-level approaches (Huntley, 1990), biomization methods (Davis et al., 2015; Prentice et al., 1996),
modern analogue technique (MAT; Zanon et al., 2018), and pseudobiomization (Fyfe et al., 2015). These approaches capture
the major trends in vegetation patterns over the course of the Holocene (Roberts et al., 2018) and biomization methods have
proved useful for evaluation of climate model results (e.g. Prentice and Webb III, 1998). The results of these forms of pollen
data manipulation either classify pollen data into discrete classes (e.g. biomization, pseudobiomization) or are semi-
quantitative, providing at best rough estimates of the relationship between forest and open-land cover (e.g. MAT). They cannot
achieve reconstructions of the cover of e.g. evergreen versus summer-green trees, or the cover of individual tree and herb taxa.
They are thus of limited use when differentiation of plant functional types (PFTs) is essential (e.g. Strandberg et al., 2014).
Moreover, forest cover over the Holocene inferred from pollen records using these approaches differs from forest cover
obtained with REVEALS (Roberts et al., 2018); these differences indicate that biases due to the non-linearity of the pollen-
vegetation relationship are not fully corrected by MAT or (pseudo -) biomization approaches.
In this paper we present the results of the second generation of REVEALS-based reconstruction of plant cover over the
Holocene in Europe, the first generation being the reconstruction published by Trondman et al. (2015). This second generation
reconstruction is, to date, the most spatially and temporally complete estimate of plant cover for Europe across the Holocene.
As with the Trondman et al. (2015) reconstruction, this new dataset is specifically designed to be used in climate modelling.
It is performed at a spatial scale of $1° \times 1°$ (ca. $100\,\mathrm{km} \times 100\,\mathrm{km}$) across 30°-75°N, 25°W-50°E (Europe and part of the Eastern
Mediterranean-Black Sea-Caspian-Corridor) (Fig. 1). The number of pollen records used (1128), the area covered and time
length (entire Holocene) are a significant advance on the results presented in Trondman et al. (2015), which used 636 pollen
records covering NW Europe (including Poland and the Czech Republic and excluding western Russia and the Mediterranean
area), and produced estimates for five time windows (in cal yr BP, hereafter abbreviated BP): 6200-5700, 4200-3700, 700-
350, 350-100 BP and 100 BP to present. Marquer et al. (2014, 2017) produced continuous REVEALS reconstructions over the
entire Holocene, however only for transects of individual sites (19 pollen records) and groups of grid cells around them.

**A**

- 1°x1° grid cell in Boreal biome
- 1°x1° grid cell in Temperare biome
- 1°x1° grid cell in Mediterranean biome
- large bog
- small bog
- large lake
- small lake

**B**

- 1°x1° grid cell with reliable number of sites
- 1°x1° grid cell with 1 small site (lake or bog) or 1 large bog


**Figure 1: Study region, A.) Site coverage. B.) Grid cell reliability dependent on number and type of pollen records.**

The 1° × 1° scale corresponds approximatively to the spatial extent of pollen-based REVEALS reconstructions as evaluated
by empirical studies in Europe (Hellman et al., 2008b; Soepboer et al., 2010): the REVEALS estimated abundances of plant



taxa (in percentage cover) correspond most closely to the plant abundances in an area of ca. 100 km x 100 km or larger (see
Li et al., 2020 for further discussion of the spatial scale of REVEALS reconstructions). This spatial scale is appropriate for
climate models that typically use spatial scales of 0.25° to 1° (Gaillard et al., 2010). REVEALS is a mechanistic model that
transforms pollen count data to produce quantitative reconstructions of regional vegetation (Sugita, 2007a). The model was
first tested and validated in southern Sweden (Hellman et al., 2008a, 2008b) and later in other parts of Europe and the world
(Soepboer et al., 2010; Sugita et al., 2010).

## 2 Methods

### 2.1 REVEALS model and parameters

Following the equation below, the REVEALS model calculates estimates of regional vegetation abundance in proportions or
percentage cover using fossil pollen counts from large lakes (Sugita, 2007a).
$$\hat{V}_i = \frac{n_{i,k} \Big/ \hat{\alpha}_i \int_R^{Z\max} g_i(z)dz}{\sum_{j=1}^{m}\left( n_{j,k} \Big/ \hat{\alpha}_j \int_R^{Z\max} g_j(z)dz \right)} = \frac{n_{i,k} \big/ \hat{\alpha}_i K_i}{\sum_{j=1}^{m}(n_{j,k}\big/\hat{\alpha}_j K_j)}$$

• $\hat{V}_i$ is the estimate of the regional vegetation abundance for taxon $i$ (proportion or percentage).
• $n_{i,k}$ is the pollen count of taxon $i$ at site $k$.
• $\hat{\alpha}_i$ is the estimate of pollen productivity (relative pollen productivity, RPP) for taxon $i$.
• $z$ is the distance between the centre of the sedimentary basin and the pollen source.
• $g_i(z)$ is the pollen dispersal/deposition function for taxon $i$ expressed as a function of distance $z$. Fall speed of pollen
(FSP), wind speed and atmospheric conditions are parameters needed to calculate this function.
• $R$ is the radius of the sedimentary basin.
• $Z_{\max}$ is the maximum distance within which most pollen originates (i.e. the maximum spatial extent of the regional
vegetation).
• $m$ is the total number of taxa included,
• $K_i = \int_R^{Zmax} g_i(z)dz$ is the "pollen dispersal-deposition coefficient" of taxon $i$ from the border of the study site
(distance from the pollen sample corresponding to the radius $R$ of the lake) to $Z_{\max}$.





The REVEALS model was developed for pollen records from large lakes and the assumptions of the model are listed in Sugita
(2007a). The model was tested and validated in Europe (Hellman et al., 2008a; Mazier et al., 2012; Soepboer et al., 2010) and
northern America (Sugita et al., 2010). Sugita (2007a) demonstrated using simulations that, in theory, the model can also be
applied on pollen records from multiple small lakes; the REVEALS estimates will however generally have larger standard
errors than those based on pollen data from large lakes. Moreover, although the application of the model on pollen data from
bogs violates the model assumption that no plants grow on the basin, REVEALS can be applied using models of pollen
dispersal and deposition for lakes or bogs. The Prentice's model (Prentice, 1985; 1988) describes deposition of pollen at a
single point in a deposition basin and is suitable for pollen records from bogs. Sugita (1993) developed the "Prentice-Sugita
model" that describes pollen deposition in a lake, i.e. on its entire surface with a subsequent mixing in the water body before
deposition at the lake bottom. The original versions of both models use the Sutton model of pollen dispersal, i.e. a Gaussian
plume model from a ground-level source under neutral atmospheric conditions (Sutton, 1953). A Lagrangian stochastic model
of dispersion has also been introduced as an alternative for the description of pollen dispersal in models of the pollen-vegetation
relationship in general, and in the REVEALS model in particular (Theuerkauf et al., 2013; 2016). It is difficult, in both theory
and practice, to eliminate the effects of pollen coming from plants growing on sedimentary basins (e.g. Poaceae and
Cyperaceae) on regional vegetation reconstruction. Previous studies have assessed the impacts of the violation of this
assumption on the REVEALS outcomes (Mazier et al., 2012; Sugita et al., 2010; Trondman et al., 2016, 2015). An empirical
study in southern Sweden (Trondman et al., 2016) indicated that REVEALS estimates based on pollen records from multiple
small sites (lakes and/or bogs) are similar to the REVEALS estimates based on pollen records from large lakes in the same
region. The results also suggested that increasing the number of pollen records significantly decreased the standard error of
the REVEALS estimates, as expected based on simulations (Sugita, 2007a). It is therefore appropriate to use pollen records
from small bogs to increase the number of pollen records included in a REVEALS reconstruction, following the protocol of
the first generation REVEALS reconstruction for Europe (Mazier et al., 2012; Trondman et al., 2015).
The inputs needed to run the REVEALS model are: original pollen counts; relative pollen productivity estimates (RPPs) and
their standard deviation; fall speed of pollen (FSP); basin type (lake or bog); size of basin (radius); maximum extent of regional
vegetation; and wind speed (m/s) and atmospheric conditions. FSP can be calculated using measurements of the pollen grains
and the Stokes' law (Gregory, 1973). RPPs of major plant taxa can be estimated using datasets of modern pollen assemblages
and related vegetation and the Extended R-Value model (Mazier et al., 2008). RPPs exist for a large number of European plant
taxa, and syntheses of FSPs and RPPs were published earlier by Broström in 2008 and Mazier in 2012 (Broström et al., 2008;
Mazier et al., 2012). The latter was used in the "first generation" REVEALS reconstruction (Trondman et al., 2015). A new
synthesis of European RPPs was performed for this "second generation" reconstruction. Preparation of data from individual
pollen records, and the values of model parameters used, are described below (sections 2.2 and 2.3).



## 2.2 Pollen records – data compilation and preparation

1128 pollen records from 29 European countries and the Eastern Mediterranean-Black Sea-Caspian-Corridor were obtained from databases and individual data contributors. The contributing databases include: the European Pollen Database (Fyfe et al., 2009; Giesecke et al., 2014); the Alpine Palynological database (ALPADABA; Institute of Plant Sciences, University of Bern; now also archived in EPD); the Czech Quaternary palynological database (PALYCZ; Kuneš et al., 2009); PALEOPYR (Lerigoleur et al., 2015); and datasets compiled within synthesis projects from the Mediterranean region (Fyfe et al., 2018; Woodbridge et al., 2018) and the Eastern Mediterranean-Black Sea-Caspian-Corridor (EMBSeCBIO project; Marinova et al., 2018) (see Fig. 1 for map, Data availability section for data location and team list for individual pollen data contributors). We followed the protocols and criteria published in Mazier et al. (2012) and Trondman et al. (2015) for selection of pollen records and application of the REVEALS model. Available pollen records were filtered based on criteria including basin type (to exclude archaeological sites and marine records) and quality of chronological control (excluding sites with poor age-depth models or fewer than three radiocarbon dates). This resulted in 1128 pollen records from lakes and bogs, both small and large. The rationale behind the use of pollen records from small sites is based on the knowledge that REVEALS estimates based on pollen records from multiple sites provide reasonable approximations of the regional cover of plant taxa (e.g. Trondman et al., 2016; see details under section 2.1 on the REVEALS model).

The taxonomy and nomenclature of pollen morphological types from the 1128 pollen records were harmonised. The pollen morphological types were then consistently assigned to one of 31 RPP taxa (Table 1; see section 2.3 for details on the RPP dataset used in this study), following the protocol outlined in Trondman et al. (2015: SI-2). This process takes into account plant morphology, biology, and ecology of the species that are included in each pollen morphological type (see Trondman et al., 2015 for examples of harmonization between pollen-morphological types and RPP taxa). In this way, RPP-harmonized pollen count data were produced for each of the 1128 pollen records. It should be noted that the EMBSeCBIO data does not contain pollen counts from cultivars, i.e. pollen from cereals and cultivated trees were deleted from the pollen records (Marinova et al., 2018). Therefore, the cover of agricultural land (represented by Cereals in this reconstruction) will always be zero in the Eastern Mediterranean-Black Sea-Caspian-Corridor in grid cells including only pollen records from EMBSeCBIO, although agriculture did occur in the region from early Neolithic.

For application of REVEALS, an age-depth model (in cal yr BP) is required for each pollen record. We used the author's original model, the model available in the contributing database or, where necessary, a new age-depth model constructed following the approach in Trondman et al. (2015). The age-depth model for each pollen record is used to aggregate RPP-harmonised pollen count data into 25 time windows across the Holocene following a standard division used in Mazier et al. (2012) and Trondman et al. (2015), later adopted by the PAGES LandCover6k working group (Gaillard et al., 2018). The first three time windows (present–100 BP (where present is the date of coring), 100-350 BP; 350-700 BP) capture the major human-induced land-cover changes since Early Middle Ages. Subsequent time windows are contiguous 500-year long intervals (e.g. 700-1200 BP, 1200-1700 BP, 1700-2200 BP, etc.) with the oldest interval representing the start of the Holocene (11200-11700



BP). The use of 500-year long time windows is motivated by the necessity to obtain sufficiently large pollen counts for reliable
REVEALS reconstructions. Since the size of the error on the REVEALS estimate partly depends on the size of the pollen
count (Sugita, 2007a), the length of the time window should be a reasonable compromise to ensure both a useful time resolution
of the reconstruction and an acceptable reliability of the REVEALS estimate of plant cover (Trondman et al., 2015).
**Table 1: Land-cover types (LCTs) and Plant Functional Types (PFTs) according to Wolf et al. (2008) and their corresponding pollen**
**morphological types. Fall speed of pollen (FSP) and the mean relative pollen productivity (RPP) estimates using the new RPP**
**synthesis (see section 2.3 and Appendices A-C for details) with their standard errors in brackets (see text for more explanations).**
**\*The FSP values of** *Quercus* **evergreen t. and Mediterranean Ericaceae according to the original study (Mazier, unpublished) are**
**0.015 and 0.051, respectively (see Appendix B, Table B.3). The value of 0.035 (FSP of** *Quercus* **deciduous t.) and 0.038 (FSP of boreal-**
**temperate Ericaceae) were used instead (see discussion in section 4.2 for explanation).**

| Land-cover types (LCTs) | PFT | PFT definition | Plant taxa/Pollen-morphological types | FSP (m/s) | PPE (SD) |
|---|---|---|---|---|---|
| Evergreen trees (ET) | TBE1 | Shade-tolerant evergreen trees | *Picea abies* | 0.056 | 5.437 (0.097) |
| | TBE2 | Shade-tolerant evergreen trees | *Abies alba* | 0.12 | 6.875 (1.442) |
| | IBE | Shade-intolerant evergreen trees | *Pinus sylvestris* | 0.031 | 6.058 (0.237) |
| | MTBE | Mediterranean shade-tolerant broadleaved evergreen trees | *Phillyrea* | 0.015 | 0.512 (0.076) |
| | | | *Pistacia* | 0.03 | 0.755 (0.201) |
| | | | *Quercus evergreen t.* | 0.035* | 11.043 (0.261) |
| | TSE | Tall shrub, evergreen | *Juniperus communis* | 0.016 | 2.07 (0.04) |
| | MTSE | Mediterranean broadleaved tall shrubs, evergreen | Ericaceae* | 0.038* | 4.265 (0.094) |
| | | | *Buxus sempervirens* | 0.032 | 1.89 (0.068) |
| Summer green trees (ST) | IBS | Shade-intolerant summer-green trees | *Alnus glutinosa* | 0.021 | 13.562 (0.293) |
| | | | *Betula* | 0.024 | 5.106 (0.303) |
| | TBS | Shade-tolerant summer-green trees | *Carpinus betulus* | 0.042 | 4.52 (0.425) |
| | | | *Carpinus orientalis* | 0.042 | 0.24 (0.07) |
| | | | *Castanea sativa* | 0.01 | 3.258 (0.059) |
| | | | *Corylus avellana* | 0.025 | 1.71 (0.1) |
| | | | *Fagus sylvatica* | 0.057 | 5.863 (0.176) |
| | | | *Fraxinus* | 0.022 | 1.044 (0.048) |
| | | | *Quercus deciduous t. *  | 0.035 | 4.537 (0.086) |
| | | | *Tilia* | 0.032 | 1.21 (0.116) |
| | | | *Ulmus* | 0.032 | 1.27 (0.05) |
| | TSD | Tall shrub, summer-green | *Salix* | 0.022 | 1.182 (0.077) |
| Open land (OL) | LSE | Low shrub, broadleaved evergreen | *Calluna vulgaris* | 0.038 | 1.085 (0.029) |
| | GL | Grassland - all herbs | *Artemisia* | 0.025 | 3.937 (0.146) |
| | | | Amaranthaceae/Chenopodiaceae | 0.019 | 4.28 (0.27) |
| | | | *Cyperaceae* | 0.035 | 0.962 (0.05) |
| | | | *Filipendula* | 0.006 | 3 (0.285) |
| | | | *Poaceae* | 0.035 | 1 (0) |
| | | | *Plantago lanceolata* | 0.029 | 2.33 (0.201) |
| | | | *Rumex acetosa-t* | 0.018 | 3.02 (0.278) |
| | AL | Agricultural land - cereals | *Cerealia-t* | 0.06 | 1.85 (0.380) |
| | | | *Secale cereale* | 0.06 | 3.99 (0.320) |




### 2.3 Model parameter setting


For the purpose of this study, a new synthesis of the RPP values available for European plant taxa was performed in 2018-
2019 based on the latest synthesis by Mazier et al. (2012) and additional RPP studies published since then (Appendix A-C). It
provides new alternative RPP datasets for the whole of Europe, including or excluding plant taxa with dominant entomophily,
with the important addition of plant taxa from the Mediterranean area (Table A1).  The location of studies included in the RPP
synthesis are shown in Fig. C1 and related information is provided in Table C1. The selection of RPP studies, RPP values
(shown in Appendix B, Tables B1 and B2) and calculation of mean RPP and their standard error (SD) for Europe are explained
in Appendix C. The synthesis includes a total of 54 taxa for which RPP values are available (Tables B1 and B2), 39 taxa from
studies in boreal and temperate Europe, and 15 taxa from studies in Mediterranean Europe of which seven include exclusively
sub-Mediterranean and Mediterranean taxa: *Buxus sempervirens*, *Carpinus orientalis*, *Castanea sativa*, Ericaceae
(Mediterranean species), *Phillyrea*, *Pistacia* and *Quercus* evergreen type. RPP values are available from both boreal/temperate
and Mediterranean Europe for seven taxa: i.e. Poaceae (reference taxon), *Acer*, *Corylus avellana*, Apiaceae, *Artemisia*,
*Plantago lanceolata* and Rubiaceae (Table B2). Table A1 presents the new RPP dataset for the 54 plant taxa and, for
comparison, the mean RPP values from Mazier et al. (2012) and from the recent synthesis by Wieczorek & Herzschuh (2020).
Moreover, comparison with the RPP values of three studies not used in our synthesis is shown in Table A2. For the REVEALS
reconstructions presented in this paper, we excluded strictly entomophilous taxa, which resulted in a total of 31 taxa (Table 1).
The excluded taxa are Compositae SF Cichoriodae, *Leucanthemum* (*Anthemis*)-t., *Potentilla*-t., *Ranunculus acris*-t., and
Rubiaceae. We included entomophilous taxa that are known to be characterised by some anemophily, e.g. *Artemisia*,
Chenopodiaceae, Rubiaceae, and *Plantago lanceolata*. We excluded plant taxa with only one RPP value except
Chenopodiaceae, *Urtica*, *Juniperus*, and *Ulmus*, and the seven exclusively sub-Mediterranean and Mediterranean taxa
mentioned above.
The FSP values (Tables 1 and A1) for boreal and temperate plant taxa were obtained from the literature (Broström et al., 2008;
Mazier et al., 2012); these values were in turn extracted from Gregory (1973) for trees, and calculated based on pollen
measurements and Stokes' law for herbs (Broström et al., 2004). FSPs for Mediterranean taxa (*Buxus sempervirens*, *Castanea*
*sativa*, Ericaceae (Mediterranean species), *Phillyrea*, *Pistacia*, and *Quercus* evergreen type) were obtained by using pollen
measurements and Stokes' law (Mazier et al., unpublished); the FSP of *Carpinus betulus* (Mazier et al., 2012) was used for
*Carpinus orientalis* (Grindean et al., 2019).
The site radius was obtained from original publications where possible. Sites in the EMBSeCBIO were classified as small
(0.01-1 km$^2$), medium (1.1-50 km$^2$) or large (50.1-500 km$^2$). These were assigned radii of 399m, 2921m and 10000 m,
respectively. Where a site's radius could not be determined from publication, it was geolocated in Google Earth and the area
of the site was measured. A radius value was extracted assuming that a site shape is circular (Mazier et al., 2012). A constant
wind speed of 3 m/s, assumed to correspond approximatively to the modern mean annual wind speed in Europe, was used
following Trondman et al. (2015). $Z_{max}$ (maximum extent of the regional vegetation) was set to 100 km. $Z_{max}$ and wind speed



influence on REVEALS estimates has been evaluated earlier in simulation and empirical studies (Gaillard et al., 2008; Mazier
et al., 2012; Sugita, 2007a). Atmospheric conditions are assumed to be neutral (Sugita, 2007a).

**2.4 Implementation of REVEALS**

REVEALS was implemented using the REVEALS function within the LRA R-package (Abraham et al., 2014; see Code
availability, section 6). The function enables the use of deposition models for bogs (Prentice's model) and lakes (Sugita's
model), and two dispersal models (a Gaussian plume model, and a Lagrangian stochastic model taken from the DISQOVER
package (Theuerkauf et al., 2016). Within this study the Gaussian plume model was applied. The REVEALS model was run
on all pollen records within each $1° \times 1°$ grid cell across Europe. The REVEALS function runs lake and bog sites separately
within each $1° \times 1°$ grid cell, and combines results (if more than one pollen record per cell) to produce a single mean cover
estimate (in proportion) and mean standard error (SE) for each taxon.  The formulation of the SE can be found in Appendix A
of Sugita (2007a). The REVEALS SE takes into account the standard deviations on the relative pollen productivities for the
individual pollen taxa (Table 1) and the number of pollen grains counted in the sample (Sugita, 2007a). The uncertainties of
the averaged REVEALS estimates of plant taxa for a grid cell are calculated using the delta method (Stuart and Ord., 1994),
and expressed as the SEs derived from the sum of the within- and between-site variations of the REVEALS results in the grid
cell. The delta method is a mathematical solution to the problem of calculating the mean of individual SEs (see Li et al., 2020,
Appendix C, for the formula and further details). Results of the REVEALS function are extracted by time window, producing
matrices of mean REVEALS estimates of cover and 25 matrices of corresponding mean SEs for each of the 31 RPP taxa
and each grid cell. The 31 RPP taxa are also assigned to 12 plant functional types (PFTs) and three land-cover types (LCTs)
(Table 1) and their mean REVEALS estimates calculated. These PFTs follow Trondman et al. (2015), with the addition of two
PFTs for Mediterranean vegetation not reconstructed in earlier studies: Mediterranean shade-tolerant broadleaved evergreen
trees (MTBE) and Mediterranean broadleaved tall shrubs, evergreen (MTSE). The mean standard errors for LCTs and PFTs
including more than one plant taxon are calculated using the delta method (Stuart. and Ord., 1994), as explained above.

**2.5 Mapping of the REVEALS estimates**

To illustrate the information that the new REVEALS reconstruction provides, we present and describe in section 3 maps of
the REVEALS estimates (% cover) and their associated SEs for the three LCTs (Fig. 2 to 4) and five taxa for eight selected
time windows: the five taxa are Cerealia-t and *Picea abies* (Fig. 5 and 6)*,* and *Calluna vulgaris,* deciduous *Quercus*, and
evergreen *Quercus* (Fig. D1-D3). The selection of the five taxa and eight time windows is motivated essentially by notable
changes in spatial distribution of these taxa through time, with higher resolution for recent times characterised by the largest
and most rapid human-induced changes in vegetation cover. For visualisation purposes the estimates are mapped in nine %
cover classes. These fractions are the same for the three LCTs (Figures 2-4), and the mapped output can therefore be directly
compared. In contrast, the colour scales used for the five taxa vary between maps depending on the abundance of the PFT/taxon
(Fig. 5 and 6, D1-D3). Different taxa thus have different scales and maps cannot be directly compared. We visualise uncertainty





in our data by plotting the SE as a circle inside each grid cell; it is the coefficient of variation (CV, i.e. the standard error
divided by the REVEALS estimate). Circles are scaled to fill the grid cell if the SE is equal or greater than the mean REVEALS
estimate (i.e. CV ≥ 1). Grid-based REVEALS results that are based on pollen records from just large bogs, or single small
bogs or lakes, provide lower quality results (see section 2.1 on the REVEALS model, and discussion section 4.1). Grid cells
for which this is the case are detailed in Table GC_quality_by_TW (see section 5, Data availability), by time window.  It
should be stressed that the percentage scale ranges we use here are different from those used in the maps of Trondman et al.,
(2015) and, therefore, the data visualisation we present cannot be directly compared with that of the 2015 study.

**3 Results**

The full results, or REVEALS dataset, include mean REVEALS values (in proportions) and their related mean SE for 31
individual tree and herb taxa, twelve PFTs and three LCTs for each grid cell in 25 consecutive time windows of the Holocene
(11.7 k BP to present) (see Data availability section). Here, results are illustrated by maps of the three LCTs (Fig. 2-4) and five
taxa (Fig. 5-6, D1-D3).  The presented maps are not part of the published dataset archived in Pangea (see Data availability,
section 5), they are examples of how the data can be presented and what they can be used for.

**3.1 Land-cover types**

The three land-cover types are evergreen trees (ET), summer-green trees (ST) and open land (OL). ET includes six PFTs which
are composed of nine pollen-morphological types (here after referred to as taxa). ST includes four PFTs which are composed
of eleven taxa while OL includes three PFTs that are in turn composed of nine taxa (Table 1).

**3.1.1 Open Land (OL)**

At the start of the Holocene, OL (Fig. 2) is higher in western Europe where it generally exceeds 80% cover, compared with
central Europe where it is more typically ~60%. There is a general decline in OL cover through the early Holocene. At 5700-
6200 BP most grid cells in central Europe have OL cover values between 10-50%. In western Europe, whilst OL is generally
reduced, several grid cells on the Atlantic fringe of northern Scotland persistently maintain 80-90% OL cover. OL increases
from the mid-Holocene, and by 2700-3200 BP the British Isles, France, Germany and the Mediterranean region have grid cells
recording OL values >70%. In central, northern and eastern Europe grid cells OL values vary between 10 - 70% at 2700-3200
BP. Time windows from the last two millennia show a consistent increase in OL with values >60% across most of central,
southern and western Europe and 20-70% in northern Europe.





% estimated regional vegetation cover

no data  0  10  20  30  40  50  60  70  80  90




**Figure 2. Grid-based REVEALS estimates of Open Land (OL) cover for eight Holocene time windows. Percentage cover of open land in 10% intervals represented by increasing shades of green. Grey cells: cells without pollen data for the time window, but with pollen data in other time windows. Circles in grid cells represent the coefficient of variation (CV; the standard error divided by the REVEALS estimate). When SE ≥ REVEALS estimate, the circle fills the entire grid cell and the REVEALS estimate is not different from zero. This occurs mainly where REVEALS estimates are low.**

### 3.1.2 Evergreen Trees (ET)

Evergreen tree (ET) cover (Fig. 3) at 9700-10200 BP is <30% across Europe, and by 7700-8200 BP fewer than 30 grid cells show ET >50%. ET percentage cover slowly increases through the early Holocene and at 5700-6200 BP groups of grid cells in southern Europe record >80%, while in northern Europe ET cover ranges between 10% and 60%. There is a consistent increase in ET cover over Europe during the mid- and late-Holocene with ET cover peaking at 2700-3200 BP before starting to reduce. Across western parts of Europe, including the British Isles, western France, Denmark, and the Netherlands ET never exceeds 20% cover.





321





**Figure 3. Grid-based REVEALS estimates of Evergreen Tress (ET) cover for eight Holocene time windows. See caption of Figure 2 for more explanations.**

### 3.1.3 Summer-green Trees (ST)

The estimate of cover of summer-green trees (Fig. 4) in the early Holocene at 9700-10200 BP is >40% across Europe. A small number (<10) of grid cells in northern, western, central and southern Europe have cover >60%. This significantly increases to 5700-6200 BP, at which time ST cover is >60% in central Europe, and 40-60% in northern Europe. ST cover remains <20% in southern Europe. From 5700-6200 BP there is a steady decline in ST cover across Europe.  At 2700-3200 BP only central Europe has ST cover >50% while the rest of Europe exhibits values <50%. There is a consistent decline over the last two millennia BP. Most of Europe has ST cover <30% in the most recent time windows (100-350 and 100 BP-present), except for a group of grid cells in the southern Baltic states and scattered records elsewhere.








**Figure 4. Grid-based REVEALS estimates of Summer-green Trees (ST) cover for eight Holocene time windows. See caption of**
**Figure 2 for more explanations.**
**3.2 Selected taxa**
In terms of PFTs, *Cerealia-t* is assigned to agricultural land (AL), *Picea abies* to shade tolerant evergreen trees (TBE1: *Picea*
*abies* is the only taxon in this PFT), *Calluna vulgaris* to low evergreen shrubs (LSE: *Calluna vulgaris* is the only taxon in this
PFT), deciduous *Quercus* to shade tolerant summer-green trees (TBS), and evergreen *Quercus* to Mediterranean shade-tolerant
broadleaved evergreen trees (MTBE) (Table 1).
**3.2.1 *Cerealia-t.***
*Cerealia-t.* (Fig. 5) is recorded at low proportions throughout the Holocene with 10-15% as the maximum cover. *Cerealia-t.*
is present in southern Europe at 9700-10200 BP with several grid cells recording >5 to 10%. Whilst such values are rare, there
are scattered grid cells in central and western Europe recording the presence of *Cerealia-t.* at very low levels (0.5-1%). These
values have high SE (greater than the REVEALS estimate) and are therefore not different from zero; they correspond to single
findings of Cerealia-t. By 5700-6200 BP, grid cells in Estonia and France record 3-5% cover, and several regions within central
and western Europe record 0-5% (0.5-1%), although with high SEs. At 2700-3200 BP, *Cerealia-t.* is recorded across central
and western Europe in the British Isles, France, Germany, and Estonia with low values. In Norway, Sweden and Finland it has
0-1% cover with high SEs. The highest cover (>5%) is observed across Europe from 1200 BP.







**Figure 5. Grid-based REVEALS estimates of Cerealia - t cover for eight Holocene time windows. Percentage cover in 0.5% intervals between 0 and 3%, 1% intervals between 3 and 5, and 5% interval between 5 and 10%. See caption of Figure 2 for more explanations.**

**3.2.2 *Picea abies***

*Picea abies* (Fig. 6) cover is low (1-2%) at 9700-10200 BP, although a number of grid cells in central and eastern Europe record values between 30 and 50%. By 7700-8200 BP, grid cells recording 30-50% cover are observed in more regions of central and eastern Europe than earlier (Russia, Estonia, Romania, Slovakia and Austria). At 5700-6200 BP, almost all central Europe has consistent but low cover of *Picea abies*; values are higher towards northeast Europe (Russia, Estonia, Latvia, Belarus and Lithuania), up to 30-50%. By 2700-3200 BP the cover of *Picea abies* has increased across central (ca. 10%) and northeast Europe (>30%). From 1200 BP, *Picea abies* is recorded in northern Europe, particularly in Norway and Sweden with some grid cells recording 25-50% cover.









**Figure 6. Grid-based REVEALS estimates of *Picea* cover for eight Holocene time windows. Percentage cover in 1% interval between 0 and 2%, 3% interval between 2 and 5%, 5% intervals between 5 and 30%, and 20% interval between 30 and 50%. See caption of Figure 2 for more explanations.**

### 3.2.3 *Calluna vulgaris*

During the Holocene, *Calluna vulgaris* cover (Fig. D1) peaks at 50%, and is largely distributed in a central European belt from the British Isles across to the southern Baltic States. At 9700-10200 BP, it is recorded in only a few grid cells, mostly in central and Western Europe, and at levels <10%. Cover slowly increases and by 7700-8200 BP, there are several grid cells with cover >25% within the British Isles, and with 10-20% cover within Denmark. At 5700-6200 BP grid cells in coastal locations in northwest Europe (particularly France, Germany and Denmark) have 50% *Calluna vulgaris* cover. Cover steadily increases within the same grid cells and by 2700-3200 BP, cover has increased in northern and Eastern Europe e.g. Norway, Estonia, with values up to 20% cover. The highest cover of *Calluna vulgaris* is recorded in the last two millennia. Although some grid cells in southeast Europe record low cover values, these have high SE.

### 3.2.4 *Quercus* deciduous

*Quercus* deciduous (Fig. D2) is recorded in central and western Europe at 9700-10200 BP at low levels (<10%), while in southern Europe (Italy) there are several grid cells recording >20% cover. By 7700-8200 BP cover in central and western Europe is between 1-10% while in northern and eastern Europe grid cells it is <2% with high SEs. During the mid-Holocene (5700-6200 BP) most of Europe, with the exception of some grid cells at the northern and southeast extremes, record *Quercus* deciduous cover values between 2-15%. By 2700-3200 BP, the cover in the same grid cells has decreased to values between 2-10%. Thereafter, the number of grid cells recording *Quercus* deciduous cover remains similar; however, the percentage cover slowly decreases and at 350-100 BP, the number of grid cells with *Quercus* deciduous cover above 5% is very low.

### 3.2.5 *Quercus* evergreen

The spatial distribution of *Quercus* evergreen (Fig. D3) remains the same throughout the Holocene. Cover of >30% is restricted to only a few grid cells and time windows. At the start of the Holocene *Quercus* evergreen is recorded with values <15% in southern Europe (Spain, Italy, Greece and Turkey) with high SEs. Cover of *Quercus* evergreen does not exceed 15% until 6700-7200 BP (not shown), in grid cells located in Turkey, Greece and Italy. From 6700-7200 BP there is an increase in the number of grid cells recording *Quercus* evergreen in southern Europe but most exhibit low cover values (<15%), and have high SEs.



## 4 Discussion

The results presented here are the first full-Holocene grid-based REVEALS estimates of land-cover change for Europe spanning the Mediterranean, temperate and boreal biomes, and highlighting the spatial and temporal dynamics of 31 taxa, 12 PFTs and 3 LCTs across Europe over the last 11700 years. Previous studies have demonstrated major differences between REVEALS results and pollen percentages (e.g. Marquer et al., 2014; Trondman et al., 2015), and it is not the scope of this paper to evaluate the results in that context. This discussion focuses on the reliability and potential of this second generation of REVEALS reconstruction for Europe for use by the wider science community.

### 4.1 Data reliability

The REVEALS results are reliant on the quality of the input datasets, namely pollen count data, chronological control for sequences, and the number and reliability of RPP estimates used (further discussion on RPPs under 4.2). The standard errors (SEs) can be considered a measure of the precision of the REVEALS results, and of reliability\quality (Trondman et al., 2015). Where SEs are equal or greater than the REVEALS estimates (represented in the maps of Fig. 2-6 and D1-D3 as a circle that fills the grid), caution should be applied in the use of the REVEALS estimates, as it implies that they are not different from zero when taking the SEs into account. Whilst this is possible within an algorithmic approach that includes estimates of uncertainty, it is conceptually impossible to have negative vegetation cover. If SEs ≥ mean REVEALS value it is therefore uncertain whether the plant taxon has cover within the grid cell. Cover may either be very low or the taxon may be absent within region (grid cell in this case).

The size of pollen counts impacts on the size of REVEALS SEs (Sugita, 2007a); larger counts result in smaller SEs. Aggregation of samples from pollen records to longer time windows results in larger count sizes and thus lower SEs (see sections 2.2 above and 4.2 below). Our input dataset includes more than 59 million individual pollen identifications, organised here into 16711 samples from 1128 sites, where a sample is an aggregated pollen count for RPP taxa for a time window at a site. 77% of samples have count sizes in excess of 1000, which is deemed most appropriate for REVEALS reconstructions (Sugita, 2007a). The mean count size across all samples is 3550. Samples with count sizes lower than 1000 are still used, but result in higher SEs. More than half of the pollen records used in the study were sourced from databases (see section 2.2). Note that the EMBSeCBIO taxonomy has been pre-standardised, and the data compilers have removed Cerealia-type. This means that for grid cells within the Eastern Mediterranean-Black Sea-Caspian-Corridor, caution is advised in the interpretation of *Cerealia-type*. Nevertheless, pollen from e.g. ruderals often related to agriculture such as *Artemisia*, Amaranthaceae/Chenopodiaceae, and *Rumex acetosa* type are included in the land-cover type open land; therefore, changes in cover of open land in the Eastern Mediterranean-Black Sea-Caspian-Corridor may be related to changes in agricultural land (see also discussion below, re agricultural, section 4.3).

Aggregation of pollen counts to time windows depends on age-depth models. We have used the best age-depth models available to us, based on the chronologies presented in Giesecke et al. (2014) for EPD sites, and through liaison with data





contributors. Nevertheless, future REVEALS runs may draw on improvements to age-depth modelling, which may result in
some original pollen count data being assigned to different time windows.
The REVEALS results here are provided for $1° \times 1°$ grid cells across Europe. The size and number of suitable pollen records
is an important factor in the quality of the REVEALS estimates for each grid cell. The REVEALS model was developed for
use with large lakes (>100-500 ha) that represent regional vegetation (Sugita, 2007a). Grid cells with multiple large lakes will
thus provide results with the highest level of certainty and reflect best the regional vegetation. These grid cell results comprising
of one or more large lakes are considered "high quality" (dark grey grids in figure 1B). It has been shown both theoretically
(Sugita, 2007a) and empirically (Fyfe et al., 2013; Trondman et al., 2016) that pollen records from multiple smaller (<100 ha)
lakes will also provide REVEALS estimates that reflect the regional vegetation. However, SEs may be larger if there is high
variability in pollen composition between records. We therefore also consider grid cells with multiple sites "high quality".
Application of REVEALS to pollen records from large bogs violates assumptions of the model (see section 2.1 above).
Therefore, REVEALS estimates for grid cells including large bogs or single small sites (lake or bog) may not be representative
of regional vegetation, particularly in areas characterised by heterogeneous vegetation. We consider such estimates as "lower
quality" (light grey grids in figure 1B), although they may still provide first-order indications on vegetation cover, and represent
an improvement on pollen percentage data (Marquer et al., 2014). Our results provide REVEALS estimates for a maximum of
420 grid cells per time window. The number and type of pollen records in a grid cell can change between time windows: not
all pollen records cover the entire Holocene. It is therefore important to consider not just the number and type of pollen records
in the total dataset, but how this changes between time windows, to assess the reliability of individual results. Results for a
maximum of 143 grid cells are based on three or more sites, 65 on two sites, and a minimum of 212 grid cells on a single site.
The results of a maximum of 67 grid cells are based on single small bogs (<400 m radius), 68 on single small lakes (<400 m
radius), and 82 on single large bogs. It implies that about half the grid cells with REVEALS results should be considered as
"lower quality" results.

**4.2 Role of RPPs and FSP in REVEALS results**

A key assumption of the REVEALS model is that RPP values are constant within the region of interest, and through time
(Sugita, 2007a). Nevertheless, it has been suggested that RPPs may vary between regions, with the variation caused by
environmental variability (climate), vegetation structure, or methodological design differences (Hellman et al., 2008a; Mazier
et al., 2012; Li et al., 2020; Wieczorek and Herzschuh, 2020). Wieczorek and Herzschuh (2020) have shown that inter-taxon
variability in RPP values is generally lower than intra-taxon variability, lending support to application of the approach we used
in the new synthesis of RPP in Europe (Appendix A-C), i.e. calculation of mean RPPs using all available RPP values that can
be considered as reliable. Nevertheless, some RPP taxa still present a challenge, for example, Ericaceae, where Mediterranean
tree forms have a greater number of inflorescences and hence may have a higher RPP than low-growth form Ericaceae in
central and northern Europe. As we are using a single RPP dataset with the RPP of Ericaceae obtained in the Mediterranean
region (more explanations below), the effect of higher pollen producing Ericaceae in the Mediterranean might result in





underrepresentation of Ericaceae cover in Central-North Europe. Unfortunately, we have only unique RPP values for Ericaceae
in both boreal-temperate Europe and Mediterranean Europe, and therefore the large difference in RPP between the two biomes
remains to be confirmed with more RPP studies.
Currently there is higher confidence in the boreal and temperate RPP values that are based on a wider set of studies increasing
the spread of values and hence reliability of the mean RPP values used (Mazier et al., 2012; Wieczorek and Herschuh, 2020),
whilst RPP values for Mediterranean taxa are based on fewer empirical RPP studies. The new RPP datasets for Europe
produced for this study (Appendix A-C) can be used in different ways. The RPPs provided in Table A1 can be used for entire
Europe, including entomophilous taxa or not, and including all values from the Mediterranean area or only the values for the
strictly sub-Mediterranean and/or Mediterranean taxa. If one uses all RPPs from the Mediterranean area, there will be taxa for
which there is both a RPP value obtained in boreal/temperate Europe and a RPP value obtained in Mediterranean Europe.
Application of both RPP values in a single REVEALS reconstruction is not straightforward to achieve, because the border
between the two regions has shifted over the Holocene. In the REVEALS reconstruction presented in this paper, we chose to
use the RPPs from Mediterranean Europe only for the sub-Mediterranean and/or Mediterranean taxa (including Ericaceae)
(Table 1 and A1), and for all other taxa we used the RPPs from boreal/ temperate Europe. The major issue with this choice is
the RPP value of Ericaceae. Using only the large value from Mediterranean Europe may lead to an under-representation of
Ericaceae (*Calluna* excluded), in particular in boreal Europe, but perhaps also in temperate Europe. Using only the small value
from boreal/temperate Europe may lead to an over-representation of Ericaceae in Mediterranean Europe.
Until we have more RPP values for each taxon, it is not possible to disentangle the effect of all factors influencing the
estimation of RPPs and to separate the effect of methodological factors from those of factors such as vegetation type, climate
and land use. The only way to evaluate the reliability of RPP datasets is to test them with modern or historical pollen
assemblages and related plant cover (Hellman et al., 2008a, 2008b). We argue that RPP values of certain taxa may not vary
substantially within some plant families or genera, while they might be variable within others, depending on the characteristics
of flowers and inflorescences that may be either very different or relatively constant within families or genera (see discussion
in (Li et al., 2018)). Therefore, we advise to use compilations of RPPs at continental or sub-continental scales rather than
compilations at multi-continental scales as the North Hemisphere dataset proposed by Wieczorek and Herzschuh (2020). We
consider the RPP selection used within this work as the most suitable for Europe to date, but expect revised and improved RPP
values as more RPP empirical studies are published. Moreover, experimentation in REVEALS applications will allow future
studies to evaluate the effects of using different RPP datasets on land-cover reconstructions (e.g. Mazier et al., 2012).
The role of FSP values in the pollen dispersal and deposition function ($g_i$ (z) in the equation of the REVEALS model, section
2.1) has been discussed by Theuerkauf et al. (2012). In this application of REVEALS we used the Gaussian Plume Model
(GPM) of dispersion and deposition as most existing RPP values have been estimated using this model. The GPM approximates
dispersal as a fast-declining curve with distance from the source plant, which implies short distances of transport for pollen
grain with high FSP compared to other models of dispersion and deposition (Theuerkauf, 2012). We have used the FSP values
obtained for *Quercus* deciduous (0.035 m/s) and boreal-temperate Ericaceae (0.037 m/s) for *Quercus* evergreen and



Mediterranean Ericaceae, respectively, although the FSP values of those two taxa were estimated to 0.015 and 0.051 in the
Mediterranean study (Table 1 and A1). The possible effect of using the lower FSP for *Quercus* evergreen (0.015 m/s) and the
high FSP for Mediterranean Ericaceae (0.051 m/s) may be lower cover of *Quercus* evergreen and higher cover of
Mediterranean Ericaceae than our results suggest. This hypothesis however requires further testing.

### 491 4.3 Use of the REVEALS results

The second generation dataset of pollen-based REVEALS land cover in Europe over the Holocene (this paper) is currently
used in two major research projects: LandClimII, and PAGES LandCover6k. LandClimII is a development of LandClimI
(Strandberg et al., 2014; Trondman et al., 2015) and studies the difference in the biogeophysical effect of land-cover change
on climate at 6000, 2500 and 200 BP (Githumbi et al., 2019). PAGES LandCover6k focuses on providing datasets on past
land-cover/land-use for climate modelling studies (Gaillard et al., 2018; Harrison et al., 2020). The first generation REVEALS
land-cover reconstruction (Marquer et al., 2014, 2017; Trondman et al., 2015) were used to evaluate other pollen-based
reconstructions of Holocene tree-cover changes in Europe (Roberts et al., 2018) and scenarios of anthropogenic land-cover
changes (ALCCs) (Kaplan et al., 2017) (see also section 1). The Trondman et al. (2015) reconstructions were used to create
continuous spatial datasets of past land cover using spatial statistical modelling (Pirzamanbein et al., 2014, 2018, 2020).
Spatially explicit datasets/maps based on the second generation of REVEALS reconstruction are currently being produced
within PAGES LandCover6k and used to evaluate and revise the HYDE (Klein Goldewijk et al., 2017) and KK10 (Kaplan et
al., 2009) ALCC scenarios. Moreover, LandCover6k archaeology-based reconstructions of past land-use change (Morrison et
al., 2021) will be integrated with the datasets of REVEALS land-cover. Besides the uses listed above, the second generation
of REVEALS reconstruction for Europe offers great potential for use in a large range of studies on past European regional
vegetation dynamics and changes in biodiversity over the Holocene (Marquer et al., 2014, 2017) and the relationship between
regional plant cover, land use, and climate over millennial and centennial time scales. Moreover the data can be used to create
all sorts of maps of plant cover that can serve in various contexts.
Several papers have discussed in depth the issues that need to be taken into account when interpreting REVEALS
reconstructions of past plant cover, in particular Trondman et al. (2015) and Marquer et al. (2017). The interpretation in terms
of human-induced vegetation change is one of the major challenges. The cover of open land (OL) may be used to assess
landscape openness, but is not a precise measure of human disturbance, as OL will include plant taxa characterizing both
naturally-open land and agricultural land that has been created by humans through the course of the Holocene with the
domestication of plants and livestock. Natural openness can occur in arctic and alpine areas, in wet regions, in river deltas and
around large lakes, as well as in eastern steppe areas. It is a particular challenge in the Mediterranean region where natural
vegetation openness represents a larger fraction of the land cover than in temperate or boreal Europe (Roberts et al., 2019).
Agricultural Land (AL) is the only PFT that includes cultivars; nevertheless, it is restricted to cereal cropping, and many other
cultivated crop types that can be identified through pollen analysis do not yet have RPP values (e.g. *Linum usitatissimum*
(common flax), *Cannabis* (hemp), *Fagopyrum* (buckwheat), beans, etc.). Moreover, the *Cerealia-t.* pollen morphological type



includes pollen from wild species of Poaceae, especially when identification relies essentially on measurements of the pollen
grain and its pore and does not consider exine structure and sculpture (Beug, 2004; Dickson, 1988).
The maps presented and described in section 3 as an illustration of the results show similar changes in spatial distributions and
quantitative cover of plant taxa and land-cover types through time, between 6000 BP and present, as the results published in
Trondman et al., (2015). The much greater potential of the new REVEALS reconstruction resides in its larger spatial extent,
covering not only boreal and temperate Europe but also southern and eastern Europe, and its contiguous time windows across
the entire Holocene, from 11700 BP to present. The quality of results is also higher in a number of grid cells in comparison to
Trondman et al (2015), where new pollen records have been included, which may in several cases decrease the standard error
on the REVEALS estimates.
**5. Data availability**
All data files reported in this work which were used for calculations, and figures are available for public download at
https://doi.pangaea.de/10.1594/PANGAEA.937075?format=html#download (Fyfe, Ralph M; Githumbi, Esther; Trondman,
Anna-Kari; Mazier, Florence; Nielsen, Anne Birgitte; Poska, Anneli; Sugita, Shinya; Woodbridge, Jessie; LandClimII
contributors; Gaillard, Marie-José (2021): A full Holocene record of transient gridded vegetation cover in Europe. PANGAEA,
https://doi.org/10.1594/PANGAEA.931856). The data and the DOI number are subject to future updates and only refer to this
version of the paper. The data available in Pangaea includes: 1) REVEALS reconstructions and their associated standard errors
for the 25 time windows; 2) Metadata of the 1128 pollen records used; 3) LandClimII contributors listing the data
contributors\collectors\databases. 4) The list of FSP and RPP values used for the reconstructions and 5) Grid cell quality
information (in terms of available pollen data, which influences the result quality: mean REVEALS estimate of plant cover)
for all grid cells.
**6. Code availability**
REVEALS was implemented using the REVEALS function within the LRA R-package (Abraham et al., 2014), available at
https://github.com/petrkunes/LRA.
Example code for data preparation and implementation of REVEALS, using two grid cells from SW Britain, is available at
https://github.com/rmfyfe/landclimII.
**7. Conclusions**
The LRA REVEALS and LOVE models (Sugita, 2007a, 2007b) are the only current land-cover reconstruction approaches
based on pollen data that incorporate assumptions that reduce the biases caused by the non-linear pollen-vegetation



relationship, differences in sedimentary archives and spatial scales. The application of the REVEALS model to 1128 pollen
records distributed across Europe has produced the first full-Holocene estimates of vegetation cover for 31 plant taxa in 1° ×
1° grid cells. These data are made available for use by the wider science community, including aggregation of results to PFTs
and LCTs. The REVEALS model assumptions are clearly stated to allow interpretation and assessment of our results and
several of the assumptions have been tested and validated. We can therefore use the land-cover reconstructions to test the role
of climate and humans on the Holocene vegetation at the regional scale in terms of changes in plant cover over time and space.
The overview of land-cover change across Europe over the Holocene can be used to track the timing and rate of vegetation
shifts which is useful in discerning the drivers of the observed change (Marquer et al., 2014; 2017). We can also study the
effect of human-induced changes in regional vegetation cover on climate, i.e. study land use as a climate forcing (e.g. Gaillard
et al., 2010; Strandberg et al., 2014; Gaillard et al., 2018; Harrison et al., 2020). Local reconstructions (LOVE) can be a
complementary approach to archaeological surveys as fine-scale human use of the landscape cannot be distinguished using
REVEALS (regional estimates). The LOVE model requires that regional plant cover is known: the REVEALS reconstructions
are therefore needed for this purpose as well, and gridded reconstructions may be a way to perform LOVE reconstructions,
although other strategies can be chosen (e.g. Cui et al., 2013; Mazier et al., 2015). Questions such as the degree of vegetation
openness though the Holocene in Europe, or on changes in the relationship between summer-green and evergreen tree cover
through time can now and in the future be answered and validated with fossil pollen data via the REVEALS approach. We
expect that in the future imprecision can be further reduced in terms of both the quality, and spatial extent, of REVEALS
estimates, as more pollen records are incorporated, and work on RPPs develops.
**Appendices**
**Appendix A - New RPP dataset for Europe**
**A.1 New synthesis of European RPPs**
Table A1 is the result of the new synthesis of RPPs available in Europe we have performed for the REVEALS reconstruction
presented in the paper. It includes RPPs for 39 plant taxa from studies in boreal and temperate Europe of which 22 (Poaceae
included) are herbs or low shrubs, and for 22 plant taxa from studies in the Mediterranean area. The two regions have RPP
values for 7 plant taxa in common. These RPPs are compared to those from two syntheses published earlier, Mazier et al.
(2012) and Wieczorek and Herzschuh (2020). The number of selected RPP values (n) for Poaceae is larger than the total
number of RPP (tn), i.e. n = tn + 1. This is due to the fact that the study of Bunting et al. 2005 does not include a value for
Poaceae and the RPP values are related to *Quercus* (Bunting et al., 2005); therefore, RPPs related to Poaceae were calculated
by assuming the RPP value for *Quercus* (related to Poaceae; *Quercus*(Poaceae)) was the same in this study region than the mean
of *Quercus*(Poaceae) RPPs from all other available studies.





The ranking of RPPs for 23 tree taxa, from the largest (13.56) to the smallest (0.240), is as follows (Poaceae included for
comparison with herbs): *Alnus*> *Quercus* evergreen (M)> *Abies alba*> *Pinus*> *Fagus sylvatica*> *Picea abies*> Ericaceae (M)>
*Betula*> *Quercus*> *Carpinus betulus*> *Populus*> *Juniperus*> *Corylus avellana*> *Castanea sativa*> *Sambucus nigra*-t.> *Ulmus*>
*Tilia*> *Salix*> *Fraxinus*> Poaceae (=1)> *Acer*> *Pistacia* (M)> *Phillyrea* (M)> *Carpinus orientalis* (M). All tree taxa have mean
RPPs larger than 1 except *Acer* (0.8), *Pistacia* (0.755), *Phillyrea* (0.512) and *Carpinus orientalis* (0.240). The ranking of RPPs
for 24 herb and low shrub taxa, from the largest (10.52) to the smallest (0.10), is as follows: *Urtica*> Chenopodiaceae> *Secale*>
*Artemisia*> Rubiaceae> *Rumex acetosa*-t.> *Filipendula*> *Plantago lanceolata*> *Trollius*> Ranunculaceae (M)> *Ranunculus*
*acris*-t.> Cerealia-t.> *Potentilla*-t.> *Plantago media*> *Calluna vulgaris*> Poaceae (=1)> Cyperaceae> *Plantago montana*>
Fabaceae (M)> Rosaceae (M)> Apiaceae> Compositae SF. Cichorioideae> *Empetrum*> *Leucanthemum* (*Anthemis*)-t.. Only
six herb taxa have RPPs larger than 3, while 12 tree taxa have RPP > than 3.
The two studies in the Mediterranean area provide single RPP values for 16 taxa, five herb taxa (Poaceae included) and 11 tree
taxa of which six are sub-Mediterranean and/or Mediterranean, and three include both temperate and Mediterranean taxa
(Cupressaceae, Ericaceae, *Fraxinus*) (Table B2). The RPP of herb taxa are significantly different between the study of
Grindean et al. (2019) and our synthesis, except for *Artemisia* (5.89 and 3, 94, respectively). The RPP of *Corylus avellana*
from the study of Mazier et al. (unpublished) (3.44) is double as large as the mean RPP in our synthesis (1.71), and the mean
RPP of *Quercus* (deciduous species) in our synthesis (4.54) is four times as large as the RPP from the study of Grindean et al.

(2019) (1.10).






**Table A1: New synthesis of European RPPs: mean RPPs with their SDs in brackets, and mean RPPs from the syntheses by Mazier et al. (2012) (St2 values) and Wieczorek and Herzschuh (2020), for comparison. This synthesis: values in bold are new mean RPPs compared to Mazier et al. (2012). The RPP values from studies in the Mediterranean area are indicated with "M" in the second column. The values emphasized in grey are the mean RPPs used in the new REVEALS reconstruction for Europe (this paper). The values of fall speed of pollen (FSP) are from Mazier et al. (2012) except those in italic, i.e. FSPs for Chenopodiaceae, *Urtica* and *Sambucus nigra*-t.** (Abraham and Kozáková, 2012)**, and *Populus* (Wieczorek and Herzschuh, 2020) and the new FSPs for Mediterranean taxa. For the three syntheses, the number of selected RPP values (n) included in the calculation of the mean RPP estimate is indicated with the total number of available RPP values (tn) in brackets. The reason why the number of selected RPP values (n) for Poaceae is larger than the total number of RPP (tn) is provided in section A1. For explanation of symbols, see captions below.**

\* Separate mean RPP values for *Calluna vulgaris*, *Empetrum*, and Ericaceae (*Calluna* and *Empetrum* excluded) in this synthesis, a single mean RPP values for all Ericales in Wieczorek and Herzschuh (2020)

\*\* Separate mean RPP values for Cerealia type (*Secale* excluded) and *Secale* in this synthesis, a single mean RPP for all cereals in Wieczorek and Herzschuh (2020)

\*\*\* Separate mean RPP values for Compositae SF Cichoriodae and *Leucanthemum* (*Anthemis*) type in this synthesis, a single mean RPP for all Asteraceae in Wieczorek and Herzschuh (2020). Note that there are no RPP for Asteraceae (Compositae SF Cichoriodae and *Leucanthemum* (*Anthemis*) type excluded) in our synthesis

^ Separate mean RPP values for *Filipendula* and *Potentilla* type in this synthesis, a single mean RPP for all Rosaceae in Wieczorek and Herzschuh (2020); note that there are no RPP for Rosaceae (*Filipendula* and *Potentilla*-t. excluded) in our synthesis; moreover *Filipendula* and *Potentilla*-t. are classified as herbs, while Rosaceae is classified as tree in Wieczorek and Herzschuh (2020)

^^ Separate mean RPP values for *Plantago lanceolata*, *P. media* and *P. montana* in this synthesis, a single mean RPP for all Plantaginaceae in Wieczorek and Herzschuh (2020); note that there are no RPP for Plantaginaceae (*Plantago lanceolata*, *P. media* and *P. montana* excluded) in our synthesis

^^^ Separate mean RPP values for *Ranunculus acris* type and *Trollius* in this synthesis, a single mean RPP for all Ranunculaceae in Wieczorek and Herzschuh (2020); note that there are no RPP for Ranunculaceae (*Ranunculus acris*-t and *Trollius* excluded) in our synthesis.



| Study | | This paper, synthesis | | | Mazier et al. 2012 St 3 | | Wieczorek & Herzschuh 2020 Europe version 2 | | |
|---|---|---|---|---|---|---|---|---|---|
| n (tn), FSP, RPP | | n (tn) | FSP | RPP (SE) | n (tn) | RPP (SE) | n(tn) | RPP (SE) | Notes |
| **HERB TAXA** | | | | | | | | | |
| **Poaceae (Reference taxon)** | | 16(15) | 0.035 | **1.00 (0.00)** | 9(8) | 1.00 (0.00) | 14(12) | 1.00 (0.00) | |
| **Herb taxa** | | | | | | | | | |
| Amaranthaceae/Chenopodiaceae | | 1(1) | *0.019* | *4.280 (0.270)* | none | none | 1(1) | 4.28 (0.27) | Same value as in this synthesis |
| Apiaceae | | 1(1) | 0.042 | 0.260 (0.010) | 1(1) | 0.26 (0.01) | 3(3) | 2.13 (0.41) | |
| Apiaceae | M | 1(1) | 0.042 | 5.910 (1.230) | | | | | |
| *Artemisia* | | 3(3) | 0.025 | **3.937 (0.146)** | 1(1) | 3.48 (0.20) | 2(2) | 4.33 (1.59) | |
| *Artemisia* | M | 1(1) | 0.014 | 5.890 (3.160) | | | | | |
| Asteraceae *Leucanth. (Anthemis)* -t*** | | 1(1) | 0.029 | 0.100 (0.010) | 1(1) | 0.10 (0.01) | | | see Asteraceae all*** |
| Asteraceae Cichorioideae*** | | 3(3) | 0.051 | 0.160 (0.020) | 3(3) | 0.16 (0.02) | 8(10) | 0.22 (0.02) | **Asteraceae all*** |
| Asteraceae Cichorioideae | M | 1(1) | 0.061 | 1.162 (0.075) | | | | | |
| Asteraceae (Asteroidae + Cichorioideae) | M | 1(1) | 0.029 | 0.160 (0.100) | | | | | |
| *Calluna vulgaris** | | 2(4) | 0.038 | 1.085 (0.029) | 2(4) | 1.09 (0.03) | | | see Ericales all* |
| Cerealia-t** | | 3(7) | 0.060 | **1.850 (0.380)** | 2(4) | 1.18 (0.04) | 4(6) | 2.36 (0.42) | **Cereals all*** |
| Cerealia-t (*Triticum* t., *Secale*, *Zea*) | M | 1(1) | 0.060 | 0.220 (0.120) | | | | | |
| Cyperaceae | | 4(6) | 0.035 | **0.962 (0.050)** | 4(6) | 0.83 (0.04) | 6(8) | 0.56 (0.02) | |
| *Empetrum** | | 1(2) | 0.038 | 0.110 (0.030) | 1(2) | 0.11 (0.03) | | | see Ericales all* |
| Ericaceae* | | 1(1) | 0.038 | 0.070 (0.040) | 1(1) | 0.07 (0.04) | 7(9) | 0.44 (0.02) | **Ericales all*** |
| Fabaceae | M | 1(1) | 0.021 | 0.400 (0.070) | | | | | |
| *Filipendula^* | | 3(3) | 0.006 | 3.000 (0.285) | 2(3) | 2.81 (0.43) | 4(6) | 0.97 (0.11) | **Rosaceae all ^** |
| *Plantago lanceolata^^* | | 4(6) | 0.029 | **2.330 (0.201)** | 3(4) | 1.04 (0.09) | 8(10) | 2.49 (0.11) | **Plantaginaceae all^^** |
| *Plantago lanceolata* | M | 1(1) | 0.029 | 0.580 (0.320) | | | | | |
| *Plantago media^^* | | 1(1) | 0.024 | 1.270 (0.180) | 1(1) | 1.27 (0.18) | | | see Plantaginaceae all^^ |
| *Plantago montana^^* | | 1(1) | 0.030 | 0.740 (0.130) | 1(1) | 0.74 (0.13) | | | see Plantaginaceae all^^ |
| *Potentilla* -t^ | | 2(3) | 0.018 | 1.720 (0.200) | 2(3) | 1.72 (0.20) | | | see Rosaceae all^ |
| Ranunculaceae | M | 1(1) | 0.020 | 2.038 (0.335) | | | | | |
| *Ranunculus acris* -t^^^ | | 2(2) | 0.014 | 1.960 (0.360) | 2(2) | 1.96 (0.36) | 3(5) | 0.99 (0.12) | **Ranunculaceae all^^^** |
| Rosaceae (*Filipend., Pot. t., Sanguisorba*) | M | 1(1) | 0.018 | 0.290 (0.120) | | | | | |
| Rubiaceae | | 2(3) | 0.019 | 3.710 (0.340) | 2(3) | 3.71 (0.34) | 3(5) | 1.56 (012) | |
| Rubiaceae | M | 1(1) | 0.019 | 0.400 (0.070) | | | | | |
| *Rumex acetosa* -t | | 3(4) | 0.018 | **3.020 (0.278)** | 3(3) | 0.85 (0.05) | 3(4) | 0.58 (0.03) | |
| *Secale* ** | | 3(3) | 0.060 | **3.990 (0.320)** | 1(1) | 3.02 (0.05) | | | see Cereals all** |
| *Trollius* ^^^ | | 1(1) | 0.013 | 2.290 (0.360) | 1(1) | 2.29 (0.36) | | | see Ranunculaceae all^^^ |
| *Urtica* | | 1(1) | *0.007* | *10.520 (0.310)* | none | none | 1(1) | 10.52 (0.31) | Same value as in this synthesis |
| **TREE TAXA** | | | | | | | | | |
| *Abies alba* | | 2(2) | 0.120 | 6.875 (1.442) | 2(2) | 6.88 (1.44) | 2(2) | 6.88 (1.44) | Same value as in this synthesis |
| *Acer* | | 2(2) | 0.056 | **0.800 (0.230)** | 2(2) | 0.80 (0.23) | 3(3) | 0.23 (0.04) | |
| *Acer* | M | 1(1) | 0.056 | **0.300 (0.090)** | | | | | |
| *Alnus* | | 5(7) | 0.021 | **13.562 (0.293)** | 3(3) | 9.07 (0.10) | 4(6) | 8.49 (0.22) | |
| *Betula* (mainly *B. pubescens*, *B. pendula*) | | 7(9) | 0.024 | **5.106 (0.303)** | 6(6) | 3.99 (0.17) | 6(8) | 4.94 (0.44) | |
| *Buxus sempervirens* | M | 1(1) | *0.032* | **1.890 (0.068)** | | | | | |
| *Carpinus betulus* | | 2(4) | 0.042 | **4.520 (0.425)** | 2(2) | 3.55 (0.43) | 3(5) | 3.09 (0.28) | |
| *Carpinus orientalis* | M | 1(1) | 0.042 | **0.240 (0.070)** | | | | | |
| *Castanea sativa* | M | 1(1) | *0.010* | **3.258 (0.059** | | | | | |
| *Corylus avellana* | | 4(4) | 0.025 | **1.710 (0.100)** | 3(3) | 1.99 (0.20) | 3(4) | 1.05 (0.33) | |
| *Corylus avellana* | M | 1(1) | 0.025 | **3.440 (0.890)** | | | | | |
| Cupressaceae (*Juniperus* 3 species) | M | 1(1) | *0.020* | **1.618 (0.161)** | | | | | See *Juniperus* |
| Ericaceae (*Arbutus unedo*, *Erica* 3 species) | M | 1(1) | *0.051* | **4.265 (0.094)** | | | | | |
| *Fagus sylvatica* | | 3(6) | 0.057 | **5.863 (0.176)** | 4(4) | 3.43 (0.09) | 3(3) | 2.35 (0.11) | |
| *Fraxinus excelsior* | | 5(6) | 0.022 | **1.044 (0.048)** | 3(3) | 1.03 (0.11) | 5(5) | 2.97 (0.25) | |
| *Fraxinus* (*F. excelsior, F. ornus*) | M | 1(1) | 0.022 | **2.990 (0.880)** | | | | | |
| *Juniperus communis* | | 1(2) | 0.016 | 2.070 (0.040) | 1(2) | 2.07 (0.04) | 1(1) | 7.94 (1.28) | |
| *Phillyrea* | M | 1(1) | *0.015* | **0.512 (0.076)** | | | | | |
| *Pistacia* | M | 1(1) | *0.030* | **0.755 (0.201)** | | | | | |
| *Picea abies* | | 4(8) | 0.056 | **5.437 (0.097)** | 4(6) | 2.62 (0.12) | 4(6) | 1.65 (0.15) | |
| *Pinus* (mainly *P. sylvestris*) | | 6(9) | 0.031 | **6.058 (0.237)** | 3(5) | 6.38 (0.45) | 4(6) | 10.86 (0.80) | |
| *Populus* | | 1(1) | *0.025* | **2.660 (1.250)** | none | none | 1(1) | 3.42 (1.60) | |
| *Quercus* (mainly *Q. robur*, *Q. petraea*) | | 6(8) | 0.035 | **4.537 (0.086)** | 4(4) | 5.83 (0.15) | 5(7) | 2.42 (0.10) | |
| *Quercus* deciduous (mainly *Q. peduncul.*) | M | 1(1) | *0.035* | **1.100 (0.350)** | | | | | |
| *Quercus* evergreen (*Q. ilex*, *Q coccifera*) | M | 1(1) | *0.015* | **11.043 (0.261)** | | | | | |
| *Salix* | | 5(5) | 0.022 | **1.182 (0.077)** | 3(4) | 1.79 (0.16) | 3(4) | 0.39 (0.06) | |
| *Sambucus nigra* -t | | 1(1) | *0.013* | **1.300 (0.120)** | none | none | 1(1) | 1.30 (0.12) | Same value as in this synthesis |
| *Tilia* | | 4(5) | 0.032 | **1.210 (0.116)** | 1(1) | 0.80 (0.03) | 3(4) | 0.93 (0.09) | |
| *Ulmus* | | 1(2) | 0.032 | **1.270 (0.050)** | 1(1) | 1.27 (0.05) | none | | |



**A.2 Comparison of the new synthesis with two earlier syntheses (Table A1)**

Of the 39 plant taxa for which we have a mean RPP in our new synthesis (N), 21 have a new mean RPP value compared to the earlier synthesis of Mazier et al. (2012) (M), 18 taxa have the same mean RPPs in both syntheses. There are three new taxa for which there were no RPP in M, i.e. Chenopodiaceae, *Sambucus nigra*-t. and *Urtica*. The mean RPPs are comparable between the two syntheses N and M, except for *Plantago lanceolata* (2.33 in N/1.04 in M), *Alnus* (13.56/9.07), *Betula* (5.11/3.09), *Carpinus betulus* (4.52/3.55), *Fagus* (5.86/3.43), *Picea* (5.44/2.62) and *Quercus* (4.54/5.83). *Abies alba* has the same RPP in all three syntheses. Chenopodiaceae, *Sambucus nigra*-t. and *Urtica* have the same single RPP values in the synthesis of Wieczorek and Herzschuh (2020) (W&H) and N. N and W&H also have comparable mean RPP values for *Artemisia*, Cereals (Cereals, *Secale* excluded in N, all Cereals in W&H), Compositae (SF Cichorioidae in N, all Asteraceae in W&H), Cyperaceae, *Plantago* (*P. lanceolata* in N, all Plantaginaceae in W&H), *Betula*, *Corylus*, *Populus* and *Tilia*. There are relatively large differences in mean RPPs in W&H and N for 16 plant taxa, although the ranking of the plant taxa in terms of their mean RPPs is almost the same. Mean RPP is larger in W&H than in N for Apiaceae (2.13/0.26), Ericales (0.44 in W&H) – *Empetrum* (0.11) and Ericaceae (0.07) in N, *Fraxinus* (2.97/1.04*), Juniperus* (7.94/2.07), *Pinus* (10.86/6.06). Mean RPP is smaller in W&H than in N for *Filipendula* (0.97/3.00), Rubiaceae (1.56/3.71), *Rumex acetosa* (0.58/2.02), *Acer* (0.23/0.80), *Alnus* (8.49/13.56), *Carpinus* (3.09/4.52), *Fagus* (2.35/5.86)), *Picea* (1.65/5.44), *Quercus* (2.42/4.54) and *Salix* (0.39/1.18).

The larger differences between the mean RPPs in N and W&H than between N and M have not been examined in detail. It is due to a slightly different selection of studies, i.e. the study of Theuerkauf et al. (2013) is not included in W&H and we did not include in N (boreal and temperate Europe, Mediterranean area excluded) the studies of Bunting et al. (2013), Kuneš et al. (2019) and Grindean et al. (2019). Another important influencing factor is the selection of RPP values for calculation of the mean RPP. Although the rules used to select RPP values are very similar between the syntheses, there are obvious differences between N and W&H that are sometimes very significant (e.g. *Juniperus*).

**A.3 Comparison of the new synthesis with three additional individual studies (Table A2)**

The RPPs from Twiddle et al. (2012) (T) for *Pinus*, *Betula* and *Calluna* are considerably larger than the mean RPPs in our synthesis (N). This is probably due to the assumption made on the RPP of *Picea* related to Poaceae. The RPP of *Picea* varies greatly between the selected studies in N, from 0.57 to 8.43 (eight values available). If we assumed that the RPP of *Picea* related to Poaceae in the study region of T was the mean RPP of the five smallest RPPs, i.e. 1.57, the RPP of the three taxa would be 4.8 for *Pinus*, 3.4 for *Betula*, and 3.3 for *Calluna*, which is more comparable to the mean RPPs in N.

Three taxa in Bunting et al. (2013) (B) have a RPP comparable to the mean RPP in N, i.e. for Cyperaceae, *Ranunculus acris*-t., and *Rumex acetosa*-t. (*R. acetosa* in B). The other taxa have a RPP in B smaller than the mean RPP in N, except *Plantago maritima* that has a larger RPP (5.8) in B than the mean RPP for *P. lanceolata* in N.

Of nine taxa, three have a RPP in Kuneš et al. (2019) (K) that is comparable to the mean RPP in N, i.e. for *Plantago lanceolata*, *Ranunculus acris*-t. and *Rumex acetosa*-t.. The other six taxa have a RPP larger than the mean RPP in N (Compositae SF





Cichorioideae, Cyperaceae and *Leucanthemum* (*Anthemis*)-t., or smaller (Chenopodiaceae, Rubiaceae) to considerably smaller
(*Urtica*). Of the 14 tree taxa, only four have a RPP in K comparable to the mean RPP in N, i.e. for *Corylus*, *Fraxinus*, *Salix*,
and *Ulmus*. For the other 10 tree taxa, the RPP in K is much smaller than the mean RPP in N for *Abies alba*, *Alnus*, *Carpinus*,
*Fagus*, *Picea*, *Pinus*, smaller for *Quercus*, and larger for *Acer* and *Tilia*.
Most of the RPP values of the three studies T, B and K are in the range of the values selected from the studies included in our
synthesis (N) except for *Urtica*, *Abies alba*, *Carpinus*, and *Pinus* in K. The Lagrangian Stochastic Model is used in K instead
of the Gaussian Plume Model in N, which may be one of the factors behind the lower RPPs in K, in particular (but not only)
for taxa with heavy pollen grains.





**Table A2: Comparison of the mean RPPs in this synthesis with the RPP estimates from Britain** (Twiddle, 2012)**,**
**Greenland** (Bunting et al., 2013) **and Czech Republic** (Kuneš et al., 2019)**. Explanations for symbols in the taxa list, see**
**caption below Table A4. + The original paper does not provide a RPP for Poaceae and values of standard deviations**
**(SDs) for the RPPs. We extracted the RPP values related to** *Picea* **from Table 5 in Twiddle et al. (2012). RPPs related**
**to Poaceae (1.00+) were then calculated by assuming that the RPP of** *Picea* **was equal to the mean RPP of** *Picea* **in**
**Europe (this synthesis) (in bold). ++ The RPPs and their SDs are not listed in the original paper, we therefore read the**
**values from Figure 4** (Bunting et al., 2013) **and the decimals are approximate. +++ Kuneš et al. (2019): we chose the**
**RPP values that were considered best by the authors, i.e. using the lake dataset (pollen from lake sediment), ERV sub-**
**model 1 and the Lagrangian Stochastic Model (for details, see Discussion section, this paper). # value for** *Plantago*
*maritima* **and ## two values for** *Rumex acetosa* **and** *Rumex acetosella***, respectively** (Bunting et al., 2013)**, for comparison**
**with** *Plantago* **spp. and** *Rumex acetosa***-t. (this paper). Underlined RPPs are close to mean RPPs (this synthesis).**

| Study<br>*Information on analysis* | This paper, synthesis<br>RPP (SE) | *Twiddle et al. (2012)+*<br>RPP - *ERV3 random GPM* | *Bunting et al. (2013)++*<br>RPP (SE) - *ERV1 GPM* | *Kunes et al (2019)+++*<br>RPP (SE) - *R ERV1 LSM* |
|---|---|---|---|---|
| **HERB TAXA** | | | | |
| **Poaceae (Reference taxon)** | 1.000 (0.000) | *1.00+* | *1.00 (0.00)* | *1.00 (0.00)* |
| **Herb taxa** | | | | |
| Amaranthaceae/Chenopodiaceae | **4.280 (0.270)** | | | *1.58 (0.74)* |
| *Calluna vulgaris** | 1.085 (0.029) | *11.42* | | |
| Comp. *Leucanthemum (Anthemis)* -t*** | 0.10 (0.01) | | | *0.94 (0.43)* |
| Comp. SF. Cichorioideae*** | 0.160 (0.020) | | | *1.04 (0.64)* |
| Cyperaceae | **0.962 (0.050)** | | *0.95 (0.05)* | *2.10 (0.88)* |
| *Plantago lanceolata^^* | **2.330 (0.201)** | | *5.8 (0.3)#* | *2.24 (0.71)* |
| *Potentilla* -t^ | 1.720 (0.200) | | *0.4 (0.03)* | |
| *Ranunculus acris* -t^^^ | 1.960 (0.360) | | *2.0 (0.1)* | *1.38 (1.13)* |
| Rubiaceae | 3.710 (0.340) | | | *1.03 (0.74)* |
| *Rumex acetosa* -t | **3.020 (0.278)** | | *3.5 (0.3)/ 2.0 (0.1)##* | *1.94 (1.35)* |
| *Urtica* | **10.520 (0.310)** | | | *1.16 (0.52)* |
| **TREE TAXA** | | | | |
| *Abies alba* | 6.875 (1.442) | | | *1.08 (0.99)* |
| *Acer* | 0.800 (0.230) | | | *1.25 (0.75)* |
| *Alnus* | **13.562 (0.293)** | | | *2.44 (0.73)* |
| *Betula* (mainly *B. pubescens , B. pendula* ) | **5.106 (0.303)** | *13.16* | *3.75 (0.4)* | *2.53 (0.91)* |
| *Carpinus betulus* | **4.520 (0.425)** | | | *1.36 (0.36)* |
| *Corylus avellana* | **1.710 (0.100)** | | | *2.31 (1.13)* |
| *Fagus sylvatica* | **5.863 (0.176)** | | | *0.88 (0.25)* |
| *Fraxinus excelsior* | **1.044 (0.048)** | | | *0.79 (0.37)* |
| *Picea abies* | **5.437 (0.097)** | *5.44* | | *2.39 (0.93)* |
| *Pinus* (mainly *P. sylvestris* ) | **6.058 (0.237)** | *16.32* | | *1.55 (0.44)* |
| *Quercus* (mainly *Q. robur , Q. petraea* ) | **4.537 (0.086)** | | | *2.08 (0.46)* |
| *Salix* | **1.182 (0.077)** | | *0.7 (0.03)* | *1.43 (0.62)* |
| *Tilia* | **1.210 (0.116)** | | | *2.30 (1.24)* |
| *Ulmus* | 1.270 (0.050) | | | *0.96 (0.77)* |






**Appendix B - Selection of RPP values and calculation of the mean RPPs and their SDs**
**B.1 Methods**
Tables B1 (Boreal and Temperate Europe) and B2 (Mediterranean Europe) list the RPP values from the 16 selected studies
according to the information on models used provided in Appendix C (Table C1) with further explanations on selection of
RPP studies. We followed similar procedures and rules as Mazier et al. (2012) and Li et al. (2018) to produce a new standard
RPP dataset for Europe. We consider that there are still too few RPP values per taxon to disentangle variability in the RPP
values for a particular taxon due to methodological issues, landscape characteristics, land use, or climate. We therefore use the
mean of selected RPP values for each taxon in the new standard RPP dataset, following Broström et al. (2008) and Mazier et
al. (2012). In boreal and temperate Europe, the number of RPP values per taxon varies between one and nine (*Betula*) (Table
B1), and in Mediterranean Europe, there is only one value per taxon (Table B2). In general, all three sub-models of the ERV
model were used in the RPP studies. We selected the RPP values obtained with the ERV sub-model considered by the authors
to have provided the best results (following the approach of Li et al., 2018). This is usually evaluated by the shape of the curve
of likelihood function scores (LFS), or log likelihood (LL) (see e.g. Twiddle et al., 2012) and the LFS and LL values
themselves.  All RPPs selected for this synthesis are expressed relative to Poaceae (RPP=1). In studies that used another
reference taxon and calculated a RPP for Poaceae, the RPPs were recalculated relative to Poaceae. In studies that did not
include a RPP value for Poaceae, it was assumed that the reference taxon had a RPP related to Poaceae equal to the mean of
the RPP values for that taxon in the other studies (e.g. Mazier et al., 2012). For simplicity, we used the value of *Quercus* (5.83)
calculated by Mazier et al. (2012) for the study by Bunting et al. (2005) (*Quercus* as reference taxon, no RPP value for
Poaceae). We could also have used the new mean RPP for *Quercus* (4.54) using our selected RPPs (five values, instead of
three in Mazier et al. (2012)). The latter would not have changed our results significantly; the mean RPP for *Quercus* would
have been 4.28 instead of 4.54 (Table A4). For the study by Baker et al. (2016), we used the RPP values obtained with Poaceae
as the reference taxon, given that the RPPs relative to *Quercus* or *Pinus* were almost identical when ERV submodel 3 was
used.  The selection of RPP values in boreal and temperate Europe for the calculation of the mean RPP values of each taxon
(values emphasized in green in Table S1.2, A and B) is based on the following rules:
1. We excluded the RPP values that were not significantly different from zero considering the lower bound of its SE,
and values that were considered as uncertain by the authors of the original publications (e.g., *Vaccinium* for Finland
(Räsänen et al., 2007), *Pinus* for Central Sweden (von Stedingk et al., 2008)). Moreover, some RPP values were
excluded as they were assumed to be outliers or unreliable based on experts' knowledge on the plants involved, the
pollen-vegetation dataset, and the field characteristics of the related studies. For example, the RPPs for Cyperaceae,
*Potentilla*-t and Rubiaceae obtained in SW Norway (Hjelle, 1998) and those for *Salix* and *Calluna vulgaris* from
Central Sweden (von Stedingk et al., 2008) were assumed to be too low compared to the values obtained in other
study areas (Mazier et al., 2012).





2.   (i) when five or more RPP estimates of pollen productivity (N≥5) were available for a pollen type, the largest and the
smallest RPP values (generally outlier values) were excluded, and the mean was calculated using the remaining three
or more RPP estimates; (ii) when N=4, the most deviating value was excluded, and the mean calculated using the
other three RPP values; (iii) when N=3, the mean was based on all values available except if one value was strongly
deviating from the other two; and (iv) when N=2, the mean was based on the two values available; an exception is
*Ulmus* for which we excluded the value from Germany (Theuerkauf et al. 2013) given that several of the RPPs in this
study are considerably higher than most values in the other available studies, i.e. for *Betula* (18.7), *Quercus* (17.85)
and *Tilia* (12.38). The latter values were also excluded from the mean RPP, as well as the unusually high values found
by Baker et al. (2016) for *Betula* (13.94), *Pinus* (23.12) and *Quercus* (18.47). Baker et al. (2016) argue that the high
RPP values might be characteristic of temperate deciduous forests that were little impacted by human activities. More
studies in this type of wooded environments would be needed to confirm this assumption. In the absence of such
studies we consider these values as outliers.

The SDs for the mean RPP values were calculated using the delta method (Stuart. and Ord., 1994), a mathematical solution to
the problem of calculating the mean of individual SDs (see e.g. Li et al. 2020 for more details).





**Table B1: Europe (Mediterranean area excluded): RPP estimates and their SDs (in brackets) with the total number of**
**taxa per study indicated and in brackets the number of taxa with selected RPP estimates. (A) Studies using moss**
**pollsters as pollen samples. (B) Studies using surface lake sediments as pollen samples. For explanation of symbols, see**
**captions below Table B1 (B).**
**(A)**

| Type of pollen sample | Moss polsters | | | | | | | |
|---|---|---|---|---|---|---|---|---|
| Region | Finland | C Sweden | S Sweden# | Norway | England## | Swiss Jura | Czech Rep* | Poland** |
| ERV submodel | ERV 3 | ERV 3 | ERV 3 | ERV 1 | ERV 1 | ERV 1 | ERV 1 | ERV 3 |
| **HERB TAXA** | | | | | | | | |
| Poaceae (Reference taxon) | 1.00 (0.00) | 1.00 (0.00) | 1.00 (0.00) | 1.00 (0.00) | 1.00 (0.00) | 1.00 (0.00) | 1.00 (0.00) | 1.00 (0.00) |
| Amaranthaceae/Chenopodiaceae | | | | | | | 4.28 (0.27) | |
| Apiaceae | | | 0.26 (0.009) | | | | | |
| *Artemisia* | | | | | | | 2.77 (0.39) | |
| *Calluna vulgaris* | | 0.30 (0.03) | 4.70 (0.69) | 1.07 (0.03) | | | | |
| Cerealia-t | | | 3.20 (1.14) | | | | 0.0462 (0.0018) | |
| Comp. *Leucanthemum* (*Anthemis* )-t | | | | 0.10 (0.008) | | | | |
| Comp. SF. Cichorioideae | | | 0.24 (0.06) | 0.06 (0.004) | | | | |
| Cyperaceae | 0.002 (0.0022) | 0.89 (0.03) | 1.00 (0.16) | 0.29 (0.01) | | 0.73 (0.08) | | |
| *Empetrum* | 0.07 (0.06) | 0.11 (0.03) | | | | | | |
| Ericaceae | | 0.07 (0.04) | | | | | | |
| *Filipendula* | | | 2.48 (0.82) | 3.39 (0.00) | | | | |
| *Plantago lanceolata* | | | 12.76 (1.83) | 1.99 (0.04) | | | 3.70 (0.77) | |
| *Plantago media* | | | | | | 1.27 (0.18) | | |
| *Plantago montana* | | | | | | 0.74 (0.13) | | |
| *Potentilla* -t | | | 2.47 (0.38) | 0.14 (0.005) | | 0.96 (0.13) | | |
| *Ranunculus acris* -t | | | 3.85 (0.72) | 0.07 (0.004) | | | | |
| Rubiaceae | | | 3.95 (0.59) | 0.42 (0.01) | | 3.47 (0.35) | | |
| *Rumex acetosa* -t | | | 4.74 (0.83) | 0.13 (0.004) | | | | |
| *Secale* | | | 3.02 (0.05) | | | | | |
| *Trollius* | | | | | | 2.29 (0.36) | | |
| *Urtica* | | | | | | | 10.52 (0.31) | |
| *Vaccinium* | 0.01 (0.01) | | | | | | | |
| **TREE TAXA** | | | | | | | | |
| *Abies* | | | | | | 3.83 (0.37) | | |
| *Acer* | | | 1.27 (0.45) | | | 0.32 (0.10) | | |
| *Alnus* | | | 4.20 (0.14) | | 8.74 (0.35) | | 2.56 (0.32) | 15.95 (0.6622) |
| *Betula* | 4.6 (0.70) | 2.24 (0.20) | 8.87 (0.13) | 6.18 (0.35) | | | | 13.94 (0.2293) |
| *Carpinus* | | | 2.53 (0.07) | | | | | 4.48 (0.0301) |
| *Corylus* | | | 1.40 (0.04) | | 1.51 (0.06) | | | 1.35 (0.0512) |
| *Fagus* | | | 6.67 (0.17) | | | 1.20 (0.16) | | |
| *Fraxinus* | | | 0.67 (0.03) | | 0.70 (0.06) | | 1.11 (0.09) | |
| *Juniperus* | | 0.11 (0.45) | 2.07 (0.04) | | | | | |
| *Picea* | | 2.78 (0.21) | 1.76 (0.00) | | | 8.43 (0.30) | | |
| *Pinus* | 8.40 (1.34) | 21.58 (2.87) | 5.66 (0.00) | | | | 6.17 (0.41) | 23.12 (0.2388) |
| *Quercus* | | | 7.53 (0.08) | | 5.83 (0.00)## | | 1.76 (0.20) | 18.47 (0.1032) |
| *Salix* | | 0.09 (0.03) | 1.27 (0.31) | | 1.05 (0.17) | | 1.19 (0.12) | |
| *Sambucus nigra* -t | | | | | | | 1.30 (0.12) | |
| *Tilia* | | | 0.80 (0.03) | | | | 1.36 (0.26) | 0.98 (0.0263) |
| *Ulmus* | | | 1.27 (0.05) | | | | | |
| Total number of taxa  39 (38) | 6 (4) | 10 (7) | 26 (25) | 12 (8) | 7 (7) | 11(10) | 13(12) | 8 (5) |





**(B)**

| Type of pollen sample | lake surface sediment | | | | |
|---|---|---|---|---|---|
| Region | Estonia | Denmark | Swiss Plateau | Germany*** | Germany **** |
| ERV submodel | ERV 3 | ERV 1 | | ERV 3 | |
| **HERB TAXA** | | | | | |
| **Poaceae (Reference taxon)** | 1.00 (0.00) | 1.00 (0.00) | 1.00 (0.00) | 1.00 (0.00) | 1.00 (0.00) |
| *Artemisia* | 3.48 (0.20) | | | | 5.56 (0.020) |
| *Calluna vulgaris* | | 1.10 (0.05) | 0.00076 (0.0019) | | |
| Cerealia-t | 1.60 (0.07) | 0.75 (0.04) | 0.17 (0.03) | 9.00 (1.92) | 0.08 (0.001) |
| Compositae *Leucanthemum* (*Anthemis*)-t | | | 0.24 (0.15) | | |
| Cyperaceae | 1.23 (0.09) | | | | |
| *Filipendula* | 3.13 (0.24) | | | | |
| *Plantago lanceolata* | | 0.90 (0.23) | | | 2.73 (0.043) |
| *Rumex acetosa* -t | | 1.56 (0.09) | | | 2.76 (0.022) |
| *Secale* | | | | 4.08 (0.96) | 4.87 (0.006) |
| **TREE TAXA** | | | 9.92 (2.86) | | |
| *Alnus* | 13.93 (0.15) | | 2.42 (0.39) | 15.51 (1.25) | 13.68 (0.049) |
| *Betula* | 1.81 (0.02) | | 4.56 (0.85) | 9.62 (1.92) | 19.70 (0.117) |
| *Carpinus* | | | 2.58 (0.39) | 9.45 (0.51) | |
| *Corylus* | | | 0.76 (0.17) | | |
| *Fagus* | | 5.09 (0.22) | 1.39 (0.21) | 5.83 (0.45) | 9.63 (0.008) |
| *Fraxinus* | | | | 6.74 (0.68) | 1.35 (0.012) |
| *Juniperus* | | | 0.57 (0.16) | | |
| *Picea* | 4.73 (0.13) | 1.19 (0.42) | 1.35 (0.45) | 1.58 (0.28) | 5.81 (0.007) |
| *Pinus* | 5.07 (0.06) | | | 5.66 (0.00) | 5.39 (0.222) |
| *Populus* | | | 2.56 (0.39) | 2.66 (1.25) | |
| *Quercus* | 7.39 (0.20) | | | 2.15 (0.17) | 17.85 (0.049) |
| *Salix* | 2.31 (0.08) | | | | |
| *Tilia* | | | | 1.47 (0.23) | 12.38 (0.101) |
| *Ulmus* | | | | | 11.51 (0.101) |
| Total number of taxa (selected values) 23 (22) | 11 (11) | 7 (7) | 13 (9) | 13 (10) | 15 (11) |


# RPPs for herbs from Broström et al. (2004); RPPs for trees from Sugita et al. (1999) (reference taxon *Juniperus*), converted
to Poaceae as reference taxon by Broström et al. (2004).
## Bunting et al. (2005), reference taxon *Quercus* and no RPP for Poaceae; RPPs relative to Poaceae calculated by Mazier et
al. (2012) assuming that the RPP of *Quercus* relative to Poaceae is the same as the mean RPP of *Quercus* from three other
studies in NW Europe.
* New RPPs from the Czech Republic (Abraham and Kozáková, 2012).
** New RPPs from Poland. Poaceae as reference taxa (see text for more details)
*** New RPPs from Germany (Matthias et al., 2012), reference taxon *Pinus*. RPPs converted to Poaceae as reference
taxon. We selected the RPP estimates obtained with the dataset of vegetation cover including only the trees that had reached
their flowering age (allFIDage) (for more information, see Matthias et al., 2012).
**** New RPPs from Germany (Theuerkauf et al., 2013); in the original publication, the ERV analysis was performed with
the Lagrangian Stochastic Model (LSM) for dispersal of pollen and with *Pinus* as reference taxon. For this synthesis, Martin



Theuerkauf redid the analysis with the Gaussian Plume Model for dispersal of pollen (Parsons and Prentice, 1981; Prentice
and Parsons, 1983) and with Poaceae as reference taxon.
**Green**: selected RPP estimates to be included in the mean RPP values.
**Red**: RPP estimates excluded because SE ≥ RPP.
**Orange**: RPP estimates excluded because of a too large difference with the other available estimates and their mean (less than
half or more than double the mean RPP).
**Light blue**: RPP estimates excluded due to its extreme high value compared to the other available estimates (much over double
the mean of the other RPPs), i.e. from the study at Bialowice forest (Poland, Baker et al., 2016) for *Betula*, *Pinus* and *Quercus*,
Central Sweden (von Stedingk et al., 2008) for *Pinus*, and Germany**** (Theuerkauf et al., 2013) for *Betula*, *Quercus*, *Tilia*,
and *Ulmus*.





**Table B2: Mediterranean area: RPP estimates and their SDs from two available studies, and mean RPPs for northern and temperate Europe (Table A1, Appendix A), for comparison. The single RPPs emphasized in green were used in the REVEALS reconstruction for Europe (this paper). The plant taxa emphasized in bold are sub-Mediterranean and/or Mediterranean plant species and genera. The values emphasized with grey shadow are the mean RPPs that were used in the REVEALS reconstruction (this paper) for entire Europe (Mediterranean area included). See Appendix B for more details. FSP values: from Mazier et al. (2012) except (') new values from Mazier et al. (unpubl.), ('') value from Abraham and Kózaková (2012), (''') value from (Commerford et al., 2013). \*, \*\*FSP from Mazier et al. (2012) used in the REVEALS reconstruction (this study) for Ericaceae (Medit)\* and *Quercus* evergreen\*\* instead of the new FSP values from Mazier et al. (unpubl.); for more explanations, see Discussion section, this paper.**

| Region | France Medit. (ERV3) | | | Roumania (ERV3) | | | Europe, Medit. excluded | | |
|---|---|---|---|---|---|---|---|---|---|
| Study reference | Mazier et al. (unpubl.) | | | Grindean et al. (2019) | | | This paper (Tables 2A, 2B) | | |
| | RPP | SD | FSP | RPP | SD | FSP | RPP | SD | FSP |
| **HERB TAXA** | | | | | | | | | |
| **Poaceae (reference taxon)** | 1.000 | 0.000 | 0.035 | 1.00 | 0.00 | 0.035 | 1.00 | 0.00 | 0.035 |
| Apiaceae | | | | 5.91 | 1.23 | 0.042 | **0.26** | **0.01** | 0.042 |
| *Artemisia* | | | | 5.89 | 3.16 | 0.014'' | **3.937** | **0.146** | **0.014''** |
| Asteraceae (Asteroideae + Cichorioideae) | | | | 0.16 | 0.10 | 0.029 | | | |
| Asteraceae Asteroidae (*Anthemis* t.., *Leucanthemum*) | | | | | | | 0.10 | 0.01 | 0.029 |
| Asteraceae Cichorioideae | 1.162 | 0.675 | 0.061' | | | | 0.16 | 0.02 | 0.05 |
| Cerealia (Cerealia t. + *Triticum* t. + *Secale* + *Zea*) | | | | 0.22 | 0.12 | 0.060 | | | |
| Cerealia (Cerealia t., *Secale* excluded) | | | | | | | **1.85** | **0.38** | **0.060** |
| Cerealia - *Secale cereale* | | | | | | | **3.99** | **0.33** | **0.060** |
| Fabaceae | | | | 0.40 | 0.07 | 0.021''' | | | |
| *Plantago lanceolata* | | | | 0.58 | 0.32 | 0.029 | **2.33** | **0.20** | **0.029** |
| Ranunculaceae | 2.038 | 0.335 | 0.020' | | | | | | |
| Ranunculaceae - *Ranunculus acris* t. | | | | | | | 1.96 | 0.36 | 0.014 |
| Ranunculaceae - *Trollius* | | | | | | | 2.29 | 0.36 | 0.013 |
| Rosaceae (*Filipendula*, *Potentilla* t., *Sanguisorba*) | | | | 0.29 | 0.12 | 0.018 | | | |
| Rosaceae - *Filipendula* | | | | | | | 3.00 | 0.28 | 0.006 |
| Rosaceae - *Potentilla* t. | | | | | | | 1.72 | 0.20 | 0.018 |
| Rubiaceae | | | | 0.40 | 0.07 | 0.019 | 3.71 | 0.34 | 0.019 |
| **TREE/SHRUB TAXA** | | | | | | | | | |
| *Acer* | | | | 0.30 | 0.09 | 0.056 | 0.80 | 0.23 | 0.056 |
| ***Buxus sempervirens*** | **1.890** | **0.068** | **0.032'** | | | | | | |
| ***Carpinus betulus*** | | | | | | | **4.52** | **0.43** | **0.042** |
| ***Carpinus orientalis*** | | | | **0.24** | **0.07** | **0.042** | | | |
| ***Castanea sativa*** | **3.258** | **0.059** | **0.010'** | | | | | | |
| *Corylus avellana* | 3.440 | 0.890 | 0.025 | | | | **1.71** | **0.10** | **0.025** |
| Cupressaceae (*Juniperus communis*, ***J. phoenica, J. oxycedrus***) | **1.618** | **0.161** | **0.020'** | | | | | | |
| Cupressaceae - *Juniperus communis* | | | | | | | **2.07** | **0.04** | **0.016** |
| Ericaceae (***Arbutus unedo, Erica arborea***, *E. cinerea*, ***E. multiflora***) | **4.265** | **0.094** | **0.051'** | | | | | | |
| Ericaceae (*Vaccinium* dominant, *Calluna* excluded) | | | | | | | 0.07 | 0.04 | **0.038\*** |
| *Fraxinus excelsior* | | | | | | | **1.04** | **0.02** | **0.022** |
| *Fraxinus* (*F. excelsior*, ***F. ornus***) | | | | 2.99 | 0.88 | 0.022 | | | |
| ***Phillyrea*** | **0.512** | **0.076** | **0.015'** | | | | | | |
| ***Pistacia*** | **0.755** | **0.201** | **0.030'** | | | | | | |
| ***Quercus* evergreen (*Q. ilex*, *Q. coccifera*)** | **11.043** | **0.261** | **0.015'** | | | | | | |
| *Quercus* deciduous (*Q.* spp, *Q. peduncularis* dominant) | | | | 1.10 | 0.35 | 0.035 | | | |
| *Quercus* deciduous (*Q. petraea* + *Q. rubra*) | | | | | | | **4.54** | **0.09** | **0.035\*\*** |
| Total number of taxa | 11 | | | 13 | | | | | |





## Appendix C - Selection of RPP studies

### C.1 Introduction

The most common method to estimate RPPs involves the application of the Extended R-Value (ERV) model on datasets of modern pollen assemblages and related vegetation cover. A summary of the ERV model and its assumptions, and an extensive description of standardised field methods for the purpose of RPP studies are found in Bunting et al. (2013). Estimation of RPPs in Europe started with the studies by Sugita et al. (1999) and Broström et al. (2004) in Southern Sweden, and Nielsen et al. (2004) in Denmark. The first tests of the RPP in pollen-based reconstructions of plant cover using the LRA's REVEALS (REgional VEgetation Abundance from Large Sites) model (Sugita, 2007a) were published by Soepboer et al. (2007) in Switzerland and Hellman et al. (2008a and b) in South Sweden. Over the last 15 years, a large number of RPP studies have been undertaken in Europe North of the Alps, but it is only recently that RPP studies were initiated in the Mediterranean area (Grindean et al., 2019; Mazier et al., unpublished). Two earlier syntheses of RPPs in Europe were published by Broström et al. (2008) and Mazier et al. (2012). From 2012 onwards, these RPP values have been used in numerous applications of the LRA's two models REVEALS and LOVE (LOcal Vegetation Estimates) (Sugita, 2007a and b) to reconstruct regional and local plant cover in Europe (Cui et al., 2013; Fyfe et al., 2013; Marquer et al., 2020; Mazier et al., 2015b; Nielsen et al., 2012; Nielsen and Odgaard, 2010; Trondman et al., 2015). Recently, Wieczorek and Herzschuh (2020) published a synthesis of the RPPs available for the Northern Hemisphere; it includes new mean RPP values for Europe that were produced independently from the synthesis we present here.

### C. 2 Selection of RPP studies and related information on methods used

The synthesis of mean RPPs presented here was produced in 2018 and applied in REVEALS reconstructions 2018-2020. Of nineteen RPP studies available (in July 2021), we selected fifteen published between 1998 and 2018 and one unpublished study in 2018 (Grindean et al., 2019). The sixteen study regions are distributed in twelve European countries (Figure C1) and detailed in Table C1. Three studies are not included in our synthesis: Britain (Twiddle et al., 2012) because of the absence of Poaceae in the calculated RPPs, curves of likelihood function scores exhibiting departures from theoretically correct curves, and doubts expressed by the authors on the reliability of the values; Greenland (Bunting et al., 2013) because this land area was not included in the REVEALS reconstruction of Holocene plant cover in Europe presented in this paper; and Czech Republic (Kuneš et al., 2019) because the study was not ready when we finalized our synthesis. However, we compare the RPP values from these three studies with the mean RPP values in this synthesis (Appendix A, Table A2).

All studies used the ERV model to calculate RPPs, and all but one study used modern pollen assemblages and vegetation; only Nielsen et al. (2004; Denmark) used historical pollen and vegetation data. Eleven studies used pollen assemblages from moss pollsters, five studies from lake sediments. Grindean et al. (2019; Romania) also used some pollen assemblages from surface

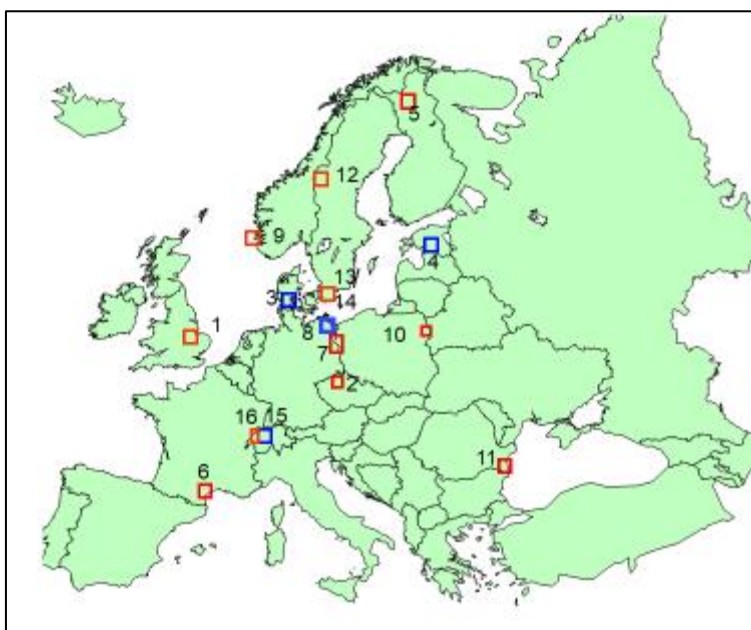

**Figure C1: Location of the selected studies of relative pollen productivities (RPP) in Europe. 1. Britain,** (Bunting et al., 2005)**; 2.**
**Czech Republic,** (Abraham and Kozáková, 2012)**; 3. Denmark,** (Nielsen, 2004)**; 4. Estonia,** (Poska et al., 2011)**; 5. Finland,** (Räsänen et
al., 2007)**; 6. France, Mazier et al. unpublished; 7. Germany,** (Matthias et al., 2012)**; 8. Germany,** (Theuerkauf et al., 2013)**; 9. Norway,**
(Hjelle, 1998)**; 10. Poland,** (Baker et al., 2016)**; 11. Romania,** (Grindean et al., 2019)**; 12. Sweden,** (von Stedingk et al., 2008)**; 13. Sweden,**
(Sugita et al., 1999)**; 14. Sweden,** (Broström et al., 2004)**; 15. Switzerland,** (Soepboer et al., 2007)**; 16. Switzerland,** (Mazier et al., 2008)**.**

soil samples. All studies used distance-weighted vegetation except two, Hjelle et al. (1998; SW Norway) and Sugita et al.
(1999; S Sweden). The Gaussian Plume Model (GPM) was used for pollen dispersal and deposition to distance-weight
vegetation, i.e. the Prentice's bog model (Parsons and Prentice, 1981; Prentice and Parsons, 1983) in studies using pollen from
moss pollsters, and the Sugita's lake model (Sugita, 1993) in studies using pollen from lake sediments (see also caption of
Table C1). In the case of the study by Theuerkauf et al. (2013), the published RPP values were calculated using the Lagrangian
Stochastic Model. For the purpose of this synthesis, Theuerkauf recalculated the RPPs using the GPM bog model in the
application of the ERV model. The distribution of sites for collection of pollen samples and vegetation data within the study
regions is random or random stratified in seven of the eleven studies using moss pollsters; the five remaining studies used
selected sites (or systematic distribution). Studies using lake sediments normally result in a systematic site distribution.
Broström et al. (2005) and Twiddle et al. (2012) showed that random distribution of sites provided better estimates of "relevant
source area of pollen" (RSAP; *sensu* Sugita, 1994) and thus of RPPs, given that the reliable RPPs are those obtained at the
RSAP distance and beyond. Both studies indicated that systematic distribution of sites have the tendency to result in curves of
likelihood function scores that do not follow the theoretical behaviour, i.e. an increase of the scores with distance until the
values reach an asymptote. However, the difference in RPPs between systematic and random sampling is generally not very
large. Nonetheless, systematic sampling may lead to uncertainty in terms of reliability of RPPs and random distribution of





sites is recommended and has generally been used in studies using moss pollsters or soil samples published from 2008 and
onwards.
**Table C1: Selection of studies for the synthesis of relative pollen productivity (RPP) estimates. Emphasized in bold:**
**additional, new studies compared to the studies included in the synthesis of Mazier et al. (2012). For explanation of**
**symbols, see captions below the Table.**

| Country | Region | No sites | Site distrib. | Pollen sample[1] | ERV sub-model | Distance weighting model[2] | Reference taxon | No taxa[3] | Reference |
|---|---|---|---|---|---|---|---|---|---|
| Britain | East Anglian: Norfolk woodlands | (34 + 19)^ | selected | M | 1 | GPM Prentice's bog | *Quercus* Poaceae** | 6 | Bunting et al. 2005 |
| **Czech Republic** | **Central Bohemia: agricultural landscape** | 54 | stratified random | M | 1 | GPM Prentice's bog | Poaceae | 13 | **Abraham & Kózaková 2012** |
| Denmark | Ancient agricultural landscape+ | 30 | selected | L++ | 1 | GPM Sugita's lake | Poaceae | 7 | Nielsen et al. 2004 |
| Estonia | Hemiboreal forest zone: mixed woodland - agricultural landscape | 40 | selected | L | 3 | GPM Sugita's lake | Poaceae | 10 | Poska et al. 2011 |
| Finland | N Finland | 24 | stratified random | M | 3 | GPM Prentice's bog | Poaceae | 6 | Räsänen et al. 2007 |
| **France** | **Mediterranean region** | 23 | random | M | 3 | GPM Prentice's bog | Poaceae | 11 | **Mazier et al. unpubl.** |
| **Germany** | **Eastern Germany: Brandenburg, agricultural landscape** | 49 | selected | L | 3 | GPM Sugita's lake | *Pinus* Poaceae* | 16 | **Matthias et al. 2012** |
| | **NE Germany: agricultural landscape** | 27 | selected | L | 3 | LSM GPM Sugita's Lake[2] | *Pinus* Poaceae* | 11 (15)[3] | **Theuerkauf et al. 2013** |
| Norway | SW Norway: Hordaland and Sogn og Fjordane, mown or grazed grass-land and heath | 39 | selected | M | 1 | None# | Poaceae | 17 | Hjelle 1998 |
| **Poland** | **NE Poland: Bialowieza Forest** | 18 | stratified random | M | 3 | GPM Prentice's bog | Poaceae | 8 | **Baker et al. 2016** |





| Romania | SE Romania: Forest-steppe region | 26 | random | M & S | 3 | GPM Prentice's bog | Poaceae | 13 | **Grindean et al. 2019** |
|---|---|---|---|---|---|---|---|---|---|
| Sweden | West- Central Sweden: Forest-tundra ecotone | 30 | random | M | 3 | GPM Prentice's bog | Poaceae | 10 | von Stedingk et al. 2008 |
| | S Sweden: ancient cultural landscapes | 114 | selected | M | 3 | None# | *Juniperus* Poaceae* | 14 (17)[3] | Sugita et al. 1999 |
| | S Sweden: unfertilized mown or grazed grasslands | 42 | selected | M | 3 | GPM Prentice's bog | Poaceae | 11 | Broström et al. 2004 |
| Switzerland | Lowland: agricultural landscape | 20 | selected | L | 3 | GPM Prentice's bog | Poaceae | 13 | Soepboer et al. 2007 |
| | Jura Mountain: pasture woodlands | 20 | (stratified) random^^ | M | 1 | GPM Prentice's bog | Poaceae | 11 | Mazier et al. 2008 |


[1] L=lakes; M=moss pollsters; S=surface soil
[2] Other distance-weighting models were used in most studies, including the Gaussian Plume Model (GPM), 1/d, 1/d$^2$
(d=distance) and the Lagrangian Stochastic Model (LSM). The GPM is used in both the model developed for bogs (Parsons
and Prentice, 1981; Prentice and Parsons, 1983) and lakes (Sugita, 1993). For this RPP synthesis, we chose the results from
the analyses using GPM rather than 1/d or 1/d$^2$. Note: In the study of Theuerkauf et al. (2013) the LSM was used. For this
synthesis, Theuerkauf recalculated his RPPs using the lake model developed by Sugita (1993).
[3] Number of plant taxa for which RPP was estimated, including the reference taxon. Note: In the study by Theuerkauf et al.
(2013) RPPs were estimated for 17 taxa using LSM. The RPPs were recalculated using the lake model (Sugita, 1993) for 15
taxa (see note under [2] above) for this synthesis. In the study of Sugita et al. (1999) RPPs were calculated for 14 trees and 3
herbs. We used only the values for the 14 trees in this synthesis, following the syntheses by Broström et al. (2008) and Mazier
et al. (2012).
^ Britain: the study includes two areas (a and b) in which RPP estimates were calculated for different sets of taxa and the two
areas have different numbers of sites: a. Calthorpe (34), 5 taxa; b. Wheatfen (17), same 5 taxa and *Corylus* (6 taxa in total).
^^ random distribution restricted to areas of the study region with existing vegetation maps (therefore no sites outside these
areas); i.e. study region including separate areas (Mazier et al., 2008).
+ Vegetation data from historical maps around 1800 CE.
++ lake sediments dated to ca. 1800.
* The reference taxon used in the original study is different from Poaceae. For this synthesis the RPPs were converted to values
relative to Poaceae.



** The study of Bunting et al. (2005) does not include a RPP for Poaceae. In order to calculate the RPPs relative to Poaceae,
it was assumed that the RPP of *Quercus* was equal to the mean of RPPs from three other studies in Europe (see Mazier et al.,
2012 for details). Although we have included new RPP values for *Quercus* in this synthesis, we did not recalculate the RPPs
from Bunting et al. (2005) with a new mean value for *Quercus*, but used the same values as in Mazier et al. (2012). For
comparison, the mean value for *Quercus* using the RPPs of the additional studies included in this synthesis is 4.28 (instead of
5.83 in Mazier et al., 2012). This would imply slightly lower RPPs in Britain also for *Alnus*, *Betula*, *Corylus*, *Fraxinus* and
*Salix*.
# no distance weighting used for vegetation data because there was no information about vegetation with increasing distance
from the pollen sample (Hjelle et al., 1998; Sugita et al., 1999). In the Swedish study, vegetation data within a $10^2$ m$^2$ (herb
taxa) and $10^3$ m$^2$ quadrat (tree taxa) centred on the pollen sample was used (Sugita et al., 1999).
**Appendix D Maps of REVEALS cover for three plant taxa (*Calluna vulgaris*, *Quercus* deciduous and *Quercus***
**evergreen)**






**Figure D1. Grid-based REVEALS estimates of *Calluna vulgaris* cover for eight Holocene time windows. Percentage cover in 2%**
**interval between 0 and 2%, 3% interval between 2 and 5%, 5% intervals between 5 – 35% and 15% interval between 35 and 50%.**
**Grey grid cells have no data (pollen) for *Calluna vulgaris* in the mapped time window. The circles represent the coefficient of**
**variation (CV; the standard error divided by the REVEALS estimate). When SE ≥ REVEALS estimate, the circle fills the entire**
**grid cell and the REVEALS estimate is not different from zero. This occurs mainly where REVEALS estimates are low.**







**Figure D2. Grid-based REVEALS estimates of *Quercus* deciduous cover in eight Holocene time windows. Percentage**
**cover in 1% interval between 0 and 2%, 3% interval between 2 and 5%, 5% intervals between 5 and 30% and 20%**
**interval between 30 and 50%. See caption of Figure A1 for more explanations.**



**Figure D3. Grid-based REVEALS estimates of *Quercus* evergreen cover for eight Holocene time windows. Percentage**
**cover in 0.5% intervals between 0 and 1%, 1% intervals between 1 and 5%, 5% intervals between 5 and 15 and 15%**
**interval between 15 and 30%. See caption of Figure A1 for more explanations.**



**+Team list**
Åkesson Christine (School of Geography & Sustainable Development, University of St. Andrews, UK), Balakauskas Lauras
(Department of Geology and Mineralogy, Vilnius University, Vilnius, Lithuania), Batalova Vlada (Lomonosov Moscow State
University, Department of Physical geography and Landscape science, Moscow, Russia),  Birks H.J.B. (Department of
Biological Sciences and Bjerknes Centre for Climate Research, University of Bergen, Norway), Bjune Anne. E. (Department
of Biological Sciences and Bjerknes Centre for Climate Research, University of Bergen, Norway), Borisova Olga (Insitute of
Geography, Russian Academy of Sciences, Moscow, Russia), Bozilova Elissaveta (Department of Botany, Sofia University
St. Kliment Ohridski, Sofia, Bulgaria), Burjachs Francesc (ICREA Barcelona, Catalonia, Spain; Rovira i Virgili University
(URV), Tarragona, Catalonia, Spain;  Institut Català de Paleoecologia Humana i Evolució Social (IPHES), Campus Sescelades
URV, W3, 43007 Tarragona, Spain), Cheddadi Rachid (Institut des Sciences de l'Evolution de Montpellier, Université de
Montpellier, CNRS-UM-IRD, Montpellier, France), Christiansen Jörg (Department of Palynology and Climate Dynamics,
Georg-August University, Göttingen, Germany), David Remi (Archeosciences Laboratory, UMR 6566 CReAAH, CNRS,
Rennes1 University, Rennes, France), de Klerk Pim (State Museum of Natural History, Karlsruhe, Germany), Dirita Federico
(Dipartimento di Biologia Ambientale, Università di Roma "La Sapienza", Piazzale Aldo Moro, 5, 00185, Roma, Italia), Döfler
Walter (Institute fur Ur- und Fruhgeschichte, Christian-Albrechts University, Kiel, Germany), Doyen Elise (Laboratoire
Chrono-Environnement, Franche-Comté University, Besançon, France), Eastwood Warren (School of Geography, Earth and
Environmental Sciences, University of Birmingham B15 2TT, UK), Etienne David (Savoie Mont Blanc University, Chambéry,
France), Feeser Ingo (Institut für Ur- und Frühgeschichte, Christian-Albrechts University, Kiel, Germany), Filipova-Marinova
Mariana (Museum of Natural History, Varna, Bulgaria), Fischer E. (Institute fur Ur- und Fruhgeschichte, Christian-Albrechts
University, Kiel, Germany), Galop Didier (GEODE UMR 5602, Toulouse University, Toulouse, France), Garcia Jose
Sebastian Carrion (Departamento de Biología Vegetal, Facultad de Biología, Universidad de Murcia, 30100 Murcia, Spain),
Herking Christa  (Institute of Botany and Landscape Ecology, EMAU, Greifswald, Germany), Herzschuh Ulrike (Alfred-
Wegener-Institut Potsdam, Germany), Jouffoy-Bapicot Isabelle (Laboratoire Chrono-Environnement, Franche-Comté
University, Besançon, France), Kasianova Alisa (Department of Palynology and Climate Dynamics, Georg-August-
University, Göttingen, Germany), Kouli Katerina (Department of Geology and Geoenvironment, National and Kapodistrian
University of Athens, Panepistimioupolis, 15784 Ilissia, Greece), Kuneš Petr (Department of Botany, Charles University,
Prague, Czech RepublicCzech), Lageras Per (The Archaeologists, National Historical Museums, Lund, Sweden), Latalowa
Malgorzata (Department of Plant Ecology, University of Gdansk, Poland), Lechterbeck Jutta (State Office for Cultural
Heritage Baden-Wuerttemberg, Germany), Leroyer Chantal (Archeosciences Laboratory, UMR 6566 CReAAH, CNRS,
Rennes1 University, Rennes, France), Leydet Michelle (European Pollen Database, IMBE, Aix-Marseille Université, Avignon
Université, IRD, Aix-en-Provence, France), Lisytstina Olga (Department of Postglacial Geology, Tallinn University of
Technology, Tallinn, Estonia), Lukanina Ekaterina (Department of Palynology and Climate Dynamics, Georg-August-





University, Göttingen, Germany), Magyari Enikő (Department of Environmental and Landscape Geography, Eötvös Loránd
University, Budapest, Hungary), Marguerie Dominique (Archeosciences Laboratory, UMR 6566 CReAAH, CNRS, Rennes1
University, Rennes, France), Mariotti Marta (Dipartimento di Biologia, Università di Firenze, Via G. La Pira, 4, 50121 Firenze,
Italy), Mensing Scott (Department of Geography, University of Nevada, Reno, NV 89557, USA), Mercuri Anna Maria
(Laboratorio di Palinologia e Paleobotanica, Dipartimento di Scienze della Vita, Università di Modena e Reggio Emilia, Italy),
Miebach Andrea (Steinmann Institute for Geology, Mineralogy, and Paleontology, University of Bonn, Bonn, Germany),
Mrotzek Almut (Institute of Botany and Landscape Ecology, EMAU ,Greifswald, Germany), Milburn Paula (College of
Science and Engineering, University of Edinburgh, Edinburgh, Scotland), Nosova Maria (Main Botanical Garden, Russian
Academy of Sciences, Moscow, Russia), Overballe-Petersen Mette (Forest & Landscape, Faculty of Life Sciences, University
of Copenhagen, Frederiksberg, Denmark),Panajiotidis Sampson (Aristotle University of Thessaloniki, Department of Forestry
and Natural Environment, PO Box: 270, GR54124 Thessaloniki, Greece), Pavlov Danail (Society of Innovative Ecologists of
Bulgaria, Varna, Bulgaria), Persson† Thomas (Department of Geology, Lund University, Lund, Sweden),  Pinke Zsolt
(Department of Physical Geography, Eötvös Loránd University, Budapest, Hungary), Ruffaldi Pascale (Laboratoire Chrono-
Environnement, Franche-Comté University, Besançon, France), Sapelko Tatyana (Institute of Limnology, Russian Academy
of Sciences , St. Petersburg, Russia), Schult Manuela (Institute of Botany and Landscape Ecology, EMAU, Greifswald,
Germany), Schmidt Monika (Department of Palynology and Climate Dynamics, Georg-August-University, Göttingen,
Germany), Stancikaite Migle  (Institute of Geology and Geography, Vilnius University, Vilnius, Lithuania), Stivrins Normunds
(Department of Geography, Faculty of Geography and Earth Sciences, University of Latvia, Jelgavas iela 1, Riga, 1004,
Latvia), Tarasov Pavel E. (Institute of Geological Sciences, Free University of Berlin, Germany), Tonkov Spassimir
(Department of Botany, Sofia University St. Kliment Ohridski, Sofia, Bulgaria), Veski Siim (Department of Geology, Tallinn
University of Technology, Tallinn, Estonia), Wick Lucia (IPNA, University of Basel, Basel, Switzerland), Wiethold Julian
(INRAP, Direction interrégionale Grand-Est Nord, Laboratoire archéobotanique, Metz, France), Woldring Henk (Groningen
Institute of Archaeology, University of Groningen, The Netherlands), Zernitskaya Valentina (Institute for Nature Management,
National Academy of Sciences of Belarusk, Minsk, Republic of Belarus).
**Author Contribution**
MJG coordinated the study as part of LandClim II and PAGES LandCover6k, two research projects for which she is the overall
coordinator and administrator. MJG, AKT, EG, FM, RF, ABN, AP and SS conceptualised the study and methodology. SS
developed the REVEALS model and helped with all issues related to the application of the model and interpretation of results.
EG, AKT, RF, FM, ABN, and AP collected new pollen records from individual authors. JW provided part of the pollen records
from the Mediterranean area (collected earlier for a separate project). MS and ST provided unpublished pollen records. EG
and AKT had the major responsibility of handling the pollen data files and collecting all related metadata. AKT collected new
values of relative pollen productivity estimates (RPPs) in Europe. MT provided unpublished RPP values for Germany and FM



for the Mediterranean area. FM, JA, VL, LM, and NNC were all involved in the unpublished RPP study in southern France, and AF, RG, ABN and IT performed the RPP study in Romania. MJG performed the selection of RPP values for the new RPP synthesis used in this paper, EG made the calculations of mean RPPs, and MJG wrote Appendices A, B, and C, and prepared the Figures and Tables therein. RF performed the REVEALS model runs and created Figure 1 and the maps of REVEALS-based plant cover (Figures 2-6 and D1-D3). EG, RF and MJG designed the manuscript, EG prepared the first draft of the manuscript and all Tables, and the final manuscript for submission, RF and MJG wrote parts of the text and edited the full manuscript. All the co-authors were involved in commenting the manuscript.

**Competing interests**

The authors declare that they have no conflict of interest.

**Figures entirely compiled by the manuscript authors**: Since such figures are part of the manuscript, they will receive the same distribution licence as the entire manuscript, namely a CC BY License. No citation is needed and no reproduction rights must be obtained.



**Acknowledgements**
This study was funded by a research project financed by the Swedish Research Council VR (Vetenskapsrådet) on
"Quantification of the bio-geophysical and biogeochemical forcings from anthropogenic de-forestation on regional Holocene
climate in Europe, LandClim II". Financial support from the Linnaeus University's Faculty of Health and Life Science is
acknowledged for Marie-José Gaillard, Anna-Kari Trondman, and Esther Githumbi. This is a contribution to the strategic
research areas MERGE (ModElling the Regional and Global Earth system) and the Past Global Change (PAGES) project and
its working group LandCover6k (http://pastglobalchanges.org/landcover6k), which in turn received support from the Swiss
National Science Foundation, the Swiss Academy of Sciences, the US National Science Foundation, and the Chinese Academy
of Sciences. Anneli Poska was supported by the ESF project number PRG323. We thank Sandy Harrison (University of
Reading, UK) for providing the pollen records from the EMBSeCBIO project.

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
