# Peer review of "European pollen-based REVEALS land-cover reconstructions for the"

_Earth System Science Data, 2021_

## Author Response (AR2)

Dear Editor,

We appreciate the time and effort taken in reviewing our manuscript and providing insightful comments. We have carefully considered the comments and tried our best to address each one of them. We hope the manuscript after careful revisions meets the expected standards. The authors welcome further constructive comments if any.

Below we provide the point-by-point responses. We do not provide a manuscript version with tracked changes because we have made numerous edits while implementing the corrections and suggestions from the reviewers and all co-authors. Instead, we indicate the main sections (lines) in which we have changed or added text in response to the reviewers' comments and requirements (see below).

We wish also to emphasize the following revisions:

- We have replaced colours by other means (symbols, emphasized borders of cells) in the Tables of Appendices A-C in order to comply with the requirements of the author guidelines. Our Tables are relatively large going over several pages and could also be formatted to be published in landscape display. We would be most thankful for advice on what is preferred.

- The Metadata Table and list of data contributors to be archived in PANGAEA have been checked carefully and corrected following the comments of co-authors and data contributors. These revised documents are now uploaded in PANGAEA.

- Finally, the language has been thoroughly edited.

Sincerely,

Esther Githumbi and co-authors

**Reviewer 1**

Pollen-based quantitative reconstruction of past vegetation and land-use is necessary for the evaluation of climate models, land-use scenarios and the study of past climate-land cover interactions. In recent years, with the help of the ERV models and the Landscape Reconstruction Algorithm (LRA, including the REVEALS and LOVE models), makes it possible to quantify plant cover from pollen records at a regional spatial scale. This study collected the original RPP values from sixteen study regions and verified the quality of RPP values; Finally, they provided a synthesized RPPs dataset of 54 taxa in Europe, and then reconstructed the history of 12 plant functional types (PFTs) and three land-cover types (LCTs) during the Holocene based on 31 selected RPP values and REVEALS model, the results provided valuable basis for climate modelling studies.

Some comments and suggestion listed below:

Q1: Why the British had sparse distribution of Evergreen Trees (such as Pinus and Picea) during the Holocene?

- **Response: The REVEALS uses the pollen counts of the taxa in the reconstruction and so the simple answer is that the sparse distribution observed is because of the low evergreen tree taxa pollen counts in the records. The vegetation historical reasons would be long to explain, but they include climate change and different processes and routes of tree migration depending on the taxa through Europe during early and mid-Holocene, and human impact during Late Holocene. *Picea* has one of the most divergent migration from its refugees S of the Alps during the last glaciation, migrating first north-eastwards up to Russia in Early Holocene to then migrate westwards towards Finland and northern Sweden in mid Holocene, to finally migrate southwards to the hemiboreal zone of Sweden in late Holocene. For more details please consult Berglund, Birks, Ralska-Jasiewiczowa, and Wright 1996: Palaeoecological events during the last 15000 years- Regional syntheses of palaeoecological**

**studies of lakes and mires in Europe. Wiley, Chichester. We did not make any revision in the manuscript related to this question as the focus of the paper is NOT on analysing the history of plant-cover change during the Holocene in Europe (which has been done elsewhere), but to present the REVEALS reconstruction and highlight its potential uses.**

line 114: What does the "reliable number of sites" mean in Figure 1B? Please provide detailed description for Figure 1.

- **Response: Reliable number of sites means grid cells that have multiple pollen records from study sites (now included in the figure caption); we explain this further in section 4.1 from line 470-489, and provide a detailed description of the figure 1B as requested (line 132-136).**

line 127: Does the "j" represent for the total number of species included for pollen proportion calculation?

- **Response: yes, more or less.** *j* stands for the 1$^{st}$ taxon of a total (maximum - *m*) number of taxa. The notation (line 155):

$$\sum_{j=1}^{m}$$

followed by an equation means in this case: sum of the results of the equation for all taxa from *j=1* (first taxon) to *m* (last taxon). The meaning of *j* and *m* is usually not provided as this is a common notation for the summation of an explicit sequence in mathematics. We let the Editor decide whether this should be explained or not.

line 296: There has only 3 PFTs which are composed 12 taxa in Summer green trees (ST); and 3 PFTs which are composed 10 taxa Open land (OL), see Table 1.
line 322: Evergreen Trees

- **Response: The mistakes have been noted and corrected.**

line 410: The mean count size across all samples is 3550. Is that right?

- **Response: Yes, most (77%) of the counts are above 1000 (line 457). Each sample is an aggregation of counts for a time window in order to obtain a large enough count for a reliable REVEALS reconstruction (lines 453-454). No modifications to the original text have been made.**

**Reviewer 2**

**General comments**
This paper concerns the use of Holocene pollen records to estimate taxon distributions and land-cover changes (LCC) through time for Europe, including the Mediterranean. It describes in detail the current state of the art use of the Landscape Reconstruction Algorithm (LRA) for the most intensely studied continental area--Europe. Of particular interest is the reduction of bias in pollen representation that allows a more accurate reconstruction of the interplay of open and forested land through time, a major proportion of which is likely related to anthropogenic land-use change.

The paper provides information on the general approach, data sources and sites, and choice of methods. It includes a detailed comparison of the numerous studies that have generated relative pollen productivity (RPP) values and a justification of the choices made in the production of the set of values used in this new iteration. The main sources of uncertainty are discussed and future improvements to methods and data are anticipated. Several useful tables

provide a comprehensive list of regional studies, RPPs (including inconsistencies among studies) and data sources. Maps provide examples of the reconstructions through time.

Overall, this is an important contribution that brings together a large body of work and makes much information accessible in one place. It is carefully constructed, comprehensive and well explained.

**Specific comments**
p 27-28. Given the likelihood that the maps created by this approach will be used in various other applications (which are indeed mentioned in the discussion), some idea of progress towards a more "believable" depiction of LCC it might be useful: i) some comments on how far the process has come, compared with simple pollen-based maps or land cover maps generated by other means, and ii) perhaps a "health warning" that the data (RPPs and maps) are still approximations, for which there are suitable but also unsuitable applications. Some mention of this is made in the discussion of the use of the LOVE package for local sites, such as might be of interest in archaeology. It is quite difficult to use this package (that is, to bring together the right site and the right data); given that, and the easy information the maps provide, it might be useful to emphasise that single grid cells on don't reflect "local" information.

- **Response: We thank the reviewer for the suggestion of making clear that the REVEALS maps are approximations of regional plant cover. We have, emphasised this more strongly by including additional text in section 4.3 (end of paragraph 1, lines 556-559), stressing that REVEALS cannot evaluate local-scale vegetation change but requires integration with the LOVE model. The advantage(s) of REVEALS reconstructions compared to pollen maps or maps based on other methods has been demonstrated elsewhere, and we have signposted this in the Introduction (lines 107-119) and at the start of the discussion section, particularly highlighting key contributions in the literature (section 4, lines 435-442). The "how far the process has come" is described in the Introduction (lines 107-119). We feel that as we do not evaluate REVEALS against other methodologies in our contribution it is not appropriate to include an extended discussion, and this would not be an original contribution here.**

P 29. It would be good if the first sentence acknowledges that while the LRA goes far beyond anything attempted before in the way of bias correction for pollen-based reconstructions, it is built on previous ideas, notably, the R-value approach of Davis, which was taken up by many other authors, including Prentice (who is mentioned) and actually, biomization, which because it uses a square-root function, does, in a non-specific way, reduce the impact of high pollen producers on large-scale reconstructions of vegetation cover.

- **Response:**
  **We agree with the reviewer that it is of importance to mention the R-Value Model of Davis and biomization in the context of this paper. We have taken into account this comment as follows:**
  **- We deleted the first sentence of the section 7 Conclusions, i.e. "**The LRA REVEALS and LOVE models (Sugita, 2007a, 2007b) are the only current land-cover reconstruction approaches based on pollen data that incorporate assumptions that reduce the biases caused by the non-linear pollen-vegetation relationship, differences in sedimentary archives and spatial scales.**". This statement is partly incorrect, as the reviewer highlighted.**
  **- We mention Davis R-Value in section 2.1 (lines 144-147) and refer to other papers for the development of pollen-vegetation modelling from the R-Value to the REVEALS model, as follows: "**The REVEALS model of Sugita (2007a) is a

generalized version of the R-Value model of Davis (1963). The development of pollen-vegetation modelling from the R-Value model, via the ERV models of Andersen (1970) and Parsons and Prentice (1981) through to the REVEALS model is described in detail in numerous earlier papers (e.g. Broström et al., 2004, Sugita, 1993, 2007a, Bunting et al., 2013)**."**

**Technical**

1. The map figures have great value, and it might be worth reviewing how easily they can be read and interpreted.
2. Fig 1: difficult to distinguish lake and bog colours
3. Figs 2-6: ensure error circles are visible on dark cell colours. It takes high magnification to see them and even then they are indistinct. Many are large and therefore important to show.

- **Response: We have done so far what we could to maximize the readability of Figures 1-6. We are also of the opinion that such Figures can be studied at the computer with zooming, which is a big advantage. Making these Figures very easily readable on paper would imply that we decrease significantly the number of Figures per Figure, which would decrease the number of maps we could include in the paper. However, we will rely on and follow the decision of the Editor in this issue and will follow it.**

4. Table B2 should read Romania (is correct in Tab C1)
5. The manuscript is well written and largely devoid of typos. I did spot a few (this list is unlikely to be comprehensive though).
   - L 147 should read The Prentice model OR Prentice's model
   - L163, no colon require after the verb "are"
   - L409 Start sentence with "Seventy-seven percent"
   - L410 no comma needed
   - L414 reword to "pollen from ruderals, for example, is often related..."
   - L513 naturally open (no hyphen)

- **Response: We have implemented all corrections of points 4 and 5**